# Maximum Likelihood Training of Score-Based Diffusion Models

**Yang Song**\*
Computer Science Department
Stanford University
yangsong@cs.stanford.edu

**Conor Durkan**\*
School of Informatics
University of Edinburgh
conor.durkan@ed.ac.uk

**Iain Murray**
School of Informatics
University of Edinburgh
i.murray@ed.ac.uk

**Stefano Ermon**
Computer Science Department
Stanford University
ermon@cs.stanford.edu

## Abstract

Score-based diffusion models synthesize samples by reversing a stochastic process that diffuses data to noise, and are trained by minimizing a weighted combination of score matching losses. The log-likelihood of score-based diffusion models can be tractably computed through a connection to continuous normalizing flows, but log-likelihood is not directly optimized by the weighted combination of score matching losses. We show that for a specific weighting scheme, the objective upper bounds the negative log-likelihood, thus enabling approximate maximum likelihood training of score-based diffusion models. We empirically observe that maximum likelihood training consistently improves the likelihood of score-based diffusion models across multiple datasets, stochastic processes, and model architectures. Our best models achieve negative log-likelihoods of 2.83 and 3.76 bits/dim on CIFAR-10 and ImageNet $32 \times 32$ without any data augmentation, on a par with state-of-the-art autoregressive models on these tasks.

## 1 Introduction

Score-based generative models [44, 45, 48] and diffusion probabilistic models [43, 19] have recently achieved state-of-the-art sample quality in a number of tasks, including image generation [48, 11], audio synthesis [5, 27, 37], and shape generation [3]. Both families of models perturb data with a sequence of noise distributions, and generate samples by learning to reverse this path from noise to data. Through stochastic calculus, these approaches can be unified into a single framework [48] which we refer to as score-based diffusion models in this paper.

The framework of score-based diffusion models [48] involves gradually diffusing the data distribution towards a given noise distribution using a stochastic differential equation (SDE), and learning the time reversal of this SDE for sample generation. Crucially, the reverse-time SDE has a closed-form expression which depends solely on the time-dependent gradient field (a.k.a., score) of the perturbed data distribution. This gradient field can be efficiently estimated by training a neural network (called a score-based model [44, 45]) with a weighted combination of score matching losses [23, 56, 46] as the objective. A key advantage of score-based diffusion models is that they can be transformed into continuous normalizing flows (CNFs) [6, 15], thus allowing tractable likelihood computation with numerical ODE solvers.

---

\*Equal contribution.

Compared to vanilla CNFs, score-based diffusion models are much more efficient to train. This is because the maximum likelihood objective for training CNFs requires running an expensive ODE solver for every optimization step, while the weighted combination of score matching losses for training score-based models does not. However, unlike maximum likelihood training, minimizing a combination of score matching losses does not necessarily lead to better likelihood values. Since better likelihoods are useful for applications including compression [21, 20, 51], semi-supervised learning [10], adversarial purification [47], and comparing against likelihood-based generative models, we seek a training objective for score-based diffusion models that is as efficient as score matching but also promotes higher likelihoods.

We show that such an objective can be readily obtained through slight modification of the weighted combination of score matching losses. Our theory reveals that with a specific choice of weighting, which we term the *likelihood weighting*, the combination of score matching losses actually upper bounds the negative log-likelihood. We further prove that this upper bound becomes tight when our score-based model corresponds to the true time-dependent gradient field of a certain reverse-time SDE. Using likelihood weighting increases the variance of our objective, which we counteract by introducing a variance reduction technique based on importance sampling. Our bound is analogous to the classic evidence lower bound used for training latent-variable models in the variational autoencoding framework [26, 39], and can be viewed as a continuous-time generalization of [43].

With our likelihood weighting, we can minimize the weighted combination of score matching losses for approximate maximum likelihood training of score-based diffusion models. Compared to weightings in previous work [48], we consistently improve likelihood values across multiple datasets, model architectures, and SDEs, with only slight degradation of Fréchet Inception distances [17]. Moreover, our upper bound on negative log-likelihood allows training with variational dequantization [18], with which we reach negative log-likelihood of **2.83** bits/dim on CIFAR-10 [28] and **3.76** bits/dim on ImageNet $32 \times 32$ [55] with no data augmentation. Our models present the first instances of normalizing flows which achieve comparable likelihood to cutting-edge autoregressive models.

## 2   Score-based diffusion models

Score-based diffusion models are deep generative models that smoothly transform data to noise using a diffusion process, and synthesize samples by learning and simulating the time-reversal of this diffusion. The overall approach is illustrated in Fig. 1.

### 2.1   Diffusing data to noise with an SDE

Let $p(\mathbf{x})$ denote the unknown distribution of a dataset consisting of $D$-dimensional i.i.d. samples. Score-based diffusion models [48] employ a stochastic differential equation (SDE) to diffuse $p(\mathbf{x})$ towards a noise distribution. The SDEs are of the form

$$\mathrm{d}\mathbf{x} = \boldsymbol{f}(\mathbf{x}, t)\,\mathrm{d}t + g(t)\,\mathrm{d}\mathbf{w}, \tag{1}$$

where $\boldsymbol{f}(\cdot, t) : \mathbb{R}^D \to \mathbb{R}^D$ is the drift coefficient, $g(t) \in \mathbb{R}$ is the diffusion coefficient, and $\mathbf{w} \in \mathbb{R}^D$ denotes a standard Wiener process (a.k.a., Brownian motion). Intuitively, we can interpret $\mathrm{d}\mathbf{w}$ as infinitesimal Gaussian noise. The solution of an SDE is a diffusion process $\{\mathbf{x}(t)\}_{t \in [0, T]}$, where $[0, T]$ is a fixed time horizon. We let $p_t(\mathbf{x})$ denote the marginal distribution of $\mathbf{x}(t)$, and $p_{0t}(\mathbf{x}' \mid \mathbf{x})$ denote the transition distribution from $\mathbf{x}(0)$ to $\mathbf{x}(t)$. Note that by definition we always have $p_0 = p$ when using an SDE to perturb the data distribution.

The role of the SDE is to smooth the data distribution by adding noise, gradually removing structure until little of the original signal remains. In the framework of score-based diffusion models, we choose $\boldsymbol{f}(\mathbf{x}, t)$, $g(t)$, and $T$ such that the diffusion process $\{\mathbf{x}(t)\}_{t \in [0, T]}$ approaches some analytically tractable prior distribution $\pi(\mathbf{x})$ at $t = T$, meaning $p_T(\mathbf{x}) \approx \pi(\mathbf{x})$. Three families of SDEs suitable for this task are outlined in [48], namely Variance Exploding (VE) SDEs, Variance Preserving (VP) SDEs, and subVP SDEs.

### 2.2   Generating samples with the reverse SDE

Sample generation in score-based diffusion models relies on time-reversal of the diffusion process. For well-behaved drift and diffusion coefficients, the forward diffusion described in Eq. (1) has an

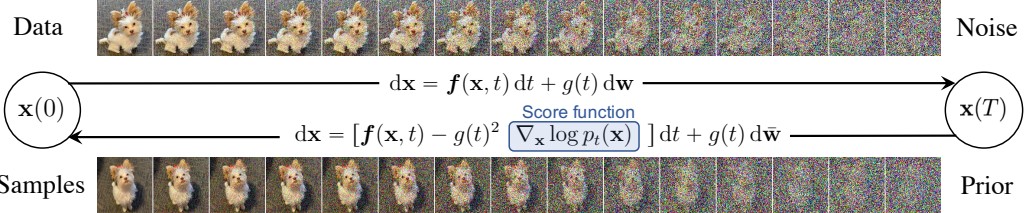

Figure 1: We can use an SDE to diffuse data to a simple noise distribution. This SDE can be reversed once we know the score of the marginal distribution at each intermediate time step, $\nabla_{\mathbf{x}} \log p_t(\mathbf{x})$.

associated reverse-time diffusion process [1, 16] given by the following SDE

$$\mathrm{d}\mathbf{x} = \left[ \boldsymbol{f}(\mathbf{x}, t) - g(t)^2 \nabla_{\mathbf{x}} \log p_t(\mathbf{x}) \right] \mathrm{d}t + g(t) \, \mathrm{d}\bar{\mathbf{w}}, \tag{2}$$

where $\bar{\mathbf{w}}$ is now a standard Wiener process in the reverse-time direction. Here $\mathrm{d}t$ represents an infinitesimal negative time step, meaning that the above SDE must be solved from $t = T$ to $t = 0$. This reverse-time SDE results in exactly the same diffusion process $\{\mathbf{x}(t)\}_{t \in [0,T]}$ as Eq. (1), assuming it is initialized with $\mathbf{x}(T) \sim p_T(\mathbf{x})$. This result allows for the construction of diffusion-based generative models, and its functional form reveals the key target for learning: the time-dependent score function $\nabla_{\mathbf{x}} \log p_t(\mathbf{x})$. Again, see Fig. 1 for a helpful visualization of this two-part formulation.

In order to estimate $\nabla_{\mathbf{x}} \log p_t(\mathbf{x})$ from a given dataset, we fit the parameters of a neural network $\boldsymbol{s}_{\boldsymbol{\theta}}(\mathbf{x}, t)$, termed a *score-based model*, such that $\boldsymbol{s}_{\boldsymbol{\theta}}(\mathbf{x}, t) \approx \nabla_{\mathbf{x}} \log p_t(\mathbf{x})$ for almost all $\mathbf{x} \in \mathbb{R}^D$ and $t \in [0, T]$. Unlike many likelihood-based generative models, a score-based model does not need to satisfy the integral constraints of a density function, and is therefore much easier to parameterize. Good score-based models should keep the following least squares loss small

$$\mathcal{J}_{\mathrm{SM}}(\boldsymbol{\theta}; \lambda(\cdot)) := \frac{1}{2} \int_0^T \mathbb{E}_{p_t(\mathbf{x})} \big[ \lambda(t) \left\| \nabla_{\mathbf{x}} \log p_t(\mathbf{x}) - \boldsymbol{s}_{\boldsymbol{\theta}}(\mathbf{x}, t) \right\|_2^2 \big] \, \mathrm{d}t, \tag{3}$$

where $\lambda \colon [0, T] \to \mathbb{R}_{>0}$ is a positive weighting function. The integrand features the well-known score matching [23] objective $\mathbb{E}_{p_t(\mathbf{x})} [\|\nabla_{\mathbf{x}} \log p_t(\mathbf{x}) - \boldsymbol{s}_{\boldsymbol{\theta}}(\mathbf{x}, t)\|_2^2]$. We therefore refer to Eq. (3) as a weighted combination of score matching losses.

With score matching techniques [56, 46], we can compute Eq. (3) up to an additive constant and minimize it for training score-based models. For example, we can use denoising score matching [56] to transform $\mathcal{J}_{\mathrm{SM}}(\boldsymbol{\theta}; \lambda(\cdot))$ into the following, which is equivalent up to a constant independent of $\boldsymbol{\theta}$:

$$\mathcal{J}_{\mathrm{DSM}}(\boldsymbol{\theta}; \lambda(\cdot)) := \frac{1}{2} \int_0^T \mathbb{E}_{p(\mathbf{x}) p_{0t}(\mathbf{x}' | \mathbf{x})} \big[ \lambda(t) \left\| \nabla_{\mathbf{x}'} \log p_{0t}(\mathbf{x}' \mid \mathbf{x}) - \boldsymbol{s}_{\boldsymbol{\theta}}(\mathbf{x}', t) \right\|_2^2 \big] \, \mathrm{d}t. \tag{4}$$

Whenever the drift coefficient $\boldsymbol{f}_{\boldsymbol{\theta}}(\mathbf{x}, t)$ is linear in $\mathbf{x}$ (which is true for all SDEs in [48]), the transition density $p_{0t}(\mathbf{x}' \mid \mathbf{x})$ is a tractable Gaussian distribution. We can form a Monte Carlo estimate of both the time integral and expectation in $\mathcal{J}_{\mathrm{DSM}}(\boldsymbol{\theta}; \lambda(\cdot))$ with a sample $(t, \mathbf{x}, \mathbf{x}')$, where $t$ is uniformly drawn from $[0, T]$, $\mathbf{x} \sim p(\mathbf{x})$ is a sample from the dataset, and $\mathbf{x}' \sim p_{0t}(\mathbf{x}' \mid \mathbf{x})$. The gradient $\nabla_{\mathbf{x}'} \log p_{0t}(\mathbf{x}' \mid \mathbf{x})$ can also be computed in closed form since $p_{0t}(\mathbf{x}' \mid \mathbf{x})$ is Gaussian.

After training a score-based model $\boldsymbol{s}_{\boldsymbol{\theta}}(\mathbf{x}, t)$ with $\mathcal{J}_{\mathrm{DSM}}(\boldsymbol{\theta}; \lambda(\cdot))$, we can plug it into the reverse-time SDE in Eq. (2). Samples are then generated by solving this reverse-time SDE with numerical SDE solvers, given an initial sample from $\pi(\mathbf{x})$ at $t = T$. Since the forward SDE Eq. (1) is designed such that $p_T(\mathbf{x}) \approx \pi(\mathbf{x})$, the reverse-time SDE will closely trace the diffusion process given by Eq. (1) in the reverse time direction, and yield an approximate data sample at $t = 0$ (as visualized in Fig. 1).

## 3 Likelihood of score-based diffusion models

The forward and backward diffusion processes in score-based diffusion models induce two probabilistic models for which we can define a likelihood. The first probabilistic model, denoted as $p_{\boldsymbol{\theta}}^{\mathrm{SDE}}(\mathbf{x})$, is given by the approximate reverse-time SDE constructed from our score-based model $\boldsymbol{s}_{\boldsymbol{\theta}}(\mathbf{x}, t)$. In particular, suppose $\{\hat{\mathbf{x}}_{\boldsymbol{\theta}}(t)\}_{t \in [0,T]}$ is a stochastic process given by

$$\mathrm{d}\hat{\mathbf{x}} = \left[ \boldsymbol{f}(\hat{\mathbf{x}}, t) - g(t)^2 \boldsymbol{s}_{\boldsymbol{\theta}}(\hat{\mathbf{x}}, t) \right] \mathrm{d}t + g(t) \, \mathrm{d}\bar{\mathbf{w}}, \quad \hat{\mathbf{x}}_{\boldsymbol{\theta}}(T) \sim \pi. \tag{5}$$

We define $p_{\boldsymbol{\theta}}^{\mathrm{SDE}}$ as the marginal distribution of $\hat{\mathbf{x}}_{\boldsymbol{\theta}}(0)$. The probabilistic model $p_{\boldsymbol{\theta}}^{\mathrm{SDE}}$ is jointly defined by the score-based model $\boldsymbol{s}_{\boldsymbol{\theta}}(\mathbf{x}, t)$, the prior $\pi$, plus the drift and diffusion coefficients of the forward SDE in Eq. (1). We can obtain a sample $\hat{\mathbf{x}}_{\boldsymbol{\theta}}(0) \sim p_{\boldsymbol{\theta}}^{\mathrm{SDE}}$ by numerically solving the reverse-time SDE in Eq. (5) with an initial noise vector $\hat{\mathbf{x}}_{\boldsymbol{\theta}}(T) \sim \pi$.

The other probabilistic model, denoted $p_{\boldsymbol{\theta}}^{\mathrm{ODE}}(\mathbf{x})$, is derived from the SDE's associated *probability flow ODE* [32, 48]. Every SDE has a corresponding probability flow ODE whose marginal distribution at each time $t$ matches that of the SDE, so that they share the same $p_t(\mathbf{x})$ for all time. In particular, the ODE corresponding to the SDE in Eq. (1) is given by

$$\frac{\mathrm{d}\mathbf{x}}{\mathrm{d}t} = \boldsymbol{f}(\mathbf{x}, t) - \frac{1}{2}g(t)^2 \nabla_{\mathbf{x}} \log p_t(\mathbf{x}). \tag{6}$$

Unlike the SDEs in Eq. (1) and Eq. (2), this ODE describes fully deterministic dynamics for the process. Notably, it still features the same time-dependent score function $\nabla_{\mathbf{x}} \log p_t(\mathbf{x})$. By approximating this score function with our model $\boldsymbol{s}_{\boldsymbol{\theta}}(\mathbf{x}, t)$, the probability flow ODE becomes

$$\frac{\mathrm{d}\tilde{\mathbf{x}}}{\mathrm{d}t} = \boldsymbol{f}(\tilde{\mathbf{x}}, t) - \frac{1}{2}g(t)^2 \boldsymbol{s}_{\boldsymbol{\theta}}(\tilde{\mathbf{x}}, t). \tag{7}$$

In fact, this ODE is an instance of a continuous normalizing flow (CNF) [15], and we can quantify how the ODE dynamics transform volumes across time in exactly the same way as these traditional flow-based models [6]. Given a prior distribution $\pi(\mathbf{x})$, and a trajectory function $\tilde{\mathbf{x}}_{\boldsymbol{\theta}} : [0, T] \to \mathbb{R}^D$ satisfying the ODE in Eq. (7), we define $p_{\boldsymbol{\theta}}^{\mathrm{ODE}}$ as the marginal distribution of $\tilde{\mathbf{x}}_{\boldsymbol{\theta}}(0)$ when $\tilde{\mathbf{x}}_{\boldsymbol{\theta}}(T) \sim \pi$. Similarly to $p_{\boldsymbol{\theta}}^{\mathrm{SDE}}$, the model $p_{\boldsymbol{\theta}}^{\mathrm{ODE}}$ is jointly defined by the score-based model $\boldsymbol{s}_{\boldsymbol{\theta}}(\mathbf{x}, t)$, the prior $\pi$, and the forward SDE in Eq. (1). Leveraging the instantaneous change-of-variables formula [6], we can evaluate $\log p_{\boldsymbol{\theta}}^{\mathrm{ODE}}(\mathbf{x})$ exactly with numerical ODE solvers. Since $p_{\boldsymbol{\theta}}^{\mathrm{ODE}}$ is a CNF, we can generate a sample $\tilde{\mathbf{x}}_{\boldsymbol{\theta}}(0) \sim p_{\boldsymbol{\theta}}^{\mathrm{ODE}}$ by numerically solving the ODE in Eq. (7) with an initial value $\tilde{\mathbf{x}}_{\boldsymbol{\theta}}(T) \sim \pi$.

Although computing $\log p_{\boldsymbol{\theta}}^{\mathrm{ODE}}(\mathbf{x})$ is tractable, training $p_{\boldsymbol{\theta}}^{\mathrm{ODE}}$ with maximum likelihood will require calling an ODE solver for every optimization step [6, 15], which can be prohibitively expensive for large-scale score-based models. Unlike $p_{\boldsymbol{\theta}}^{\mathrm{ODE}}$, we cannot evaluate $\log p_{\boldsymbol{\theta}}^{\mathrm{SDE}}(\mathbf{x})$ exactly for an arbitrary data point $\mathbf{x}$. However, we have a lower bound on $\log p_{\boldsymbol{\theta}}^{\mathrm{SDE}}(\mathbf{x})$ which allows both efficient evaluation and optimization, as will be shown in Section 4.2.

## 4 Bounding the likelihood of score-based diffusion models

Many applications benefit from models which achieve high likelihood. One example is lossless compression, where log-likelihood directly corresponds to the minimum expected number of bits needed to encode a message. Indeed, popular likelihood-based models such as variational autoencoders and normalizing flows have already found success in image compression [51, 20, 21]. Despite some known drawbacks [50], likelihood is still one of the most popular metrics for evaluating and comparing generative models.

Maximizing the likelihood of score-based diffusion models can be accomplished by either maximizing the likelihood of $p_{\boldsymbol{\theta}}^{\mathrm{SDE}}$ or $p_{\boldsymbol{\theta}}^{\mathrm{ODE}}$. Although $p_{\boldsymbol{\theta}}^{\mathrm{ODE}}$ is a continuous normalizing flow (CNF) and its log-likelihood is tractable, training with maximum likelihood is expensive. As mentioned already, it requires solving an ODE at every optimization step in order to evaluate the log-likelihood on a batch of training data. In contrast, training with the weighted combination of score matching losses is much more efficient, yet in general it does not directly promote high likelihood of either $p_{\boldsymbol{\theta}}^{\mathrm{SDE}}$ or $p_{\boldsymbol{\theta}}^{\mathrm{ODE}}$.

In what follows, we show that with a specific choice of the weighting function $\lambda(t)$, the combination of score matching losses $\mathcal{J}_{\mathrm{SM}}(\boldsymbol{\theta}; \lambda(\cdot))$ actually becomes an upper bound on $D_{\mathrm{KL}}(p \parallel p_{\boldsymbol{\theta}}^{\mathrm{SDE}})$, and can therefore serve as an efficient proxy for maximum likelihood training. In addition, we provide a related lower bound on $\log p_{\boldsymbol{\theta}}^{\mathrm{SDE}}(\mathbf{x})$ that can be evaluated efficiently on any individual datapoint $\mathbf{x}$.

### 4.1 Bounding the KL divergence with likelihood weighting

It is well-known that maximizing the log-likelihood of a probabilistic model is equivalent to minimizing the KL divergence from the data distribution to the model distribution. We show in the following theorem that for the model $p_{\boldsymbol{\theta}}^{\mathrm{SDE}}$, this KL divergence can be upper bounded by $\mathcal{J}_{\mathrm{SM}}(\boldsymbol{\theta}; \lambda(\cdot))$ when using the weighting function $\lambda(t) = g(t)^2$, where $g(t)$ is the diffusion coefficient of SDE in Eq. (1).

Table 1: SDEs and their corresponding weightings for score matching losses.

| SDE | Formula | $\lambda(t)$ in [48] | likelihood weighting |
|---|---|---|---|
| VE | $\mathrm{d}\mathbf{x} = \sigma(t)\,\mathrm{d}\mathbf{w}$ | $\sigma^2(t)$ | $\sigma^2(t)$ |
| VP | $\mathrm{d}\mathbf{x} = -\frac{1}{2}\beta(t)\mathbf{x}\,\mathrm{d}t + \sqrt{\beta(t)}\,\mathrm{d}\mathbf{w}$ | $1 - e^{-\int_0^t \beta(s)\,\mathrm{d}s}$ | $\beta(t)$ |
| subVP | $\mathrm{d}\mathbf{x} = -\frac{1}{2}\beta(t)\mathbf{x}\,\mathrm{d}t + \sqrt{\beta(t)(1 - e^{-2\int_0^t \beta(s)\,\mathrm{d}s})}\,\mathrm{d}\mathbf{w}$ | $(1 - e^{-\int_0^t \beta(s)\,\mathrm{d}s})^2$ | $\beta(t)(1 - e^{-2\int_0^t \beta(s)\,\mathrm{d}s})$ |

**Theorem 1.** *Let $p(\mathbf{x})$ be the data distribution, $\pi(\mathbf{x})$ be a known prior distribution, and $p_{\boldsymbol{\theta}}^{\mathrm{SDE}}$ be defined as in Section 3. Suppose $\{\mathbf{x}(t)\}_{t\in[0,T]}$ is a stochastic process defined by the SDE in Eq. (1) with $\mathbf{x}(0) \sim p$, where the marginal distribution of $\mathbf{x}(t)$ is denoted as $p_t$. Under some regularity conditions detailed in Appendix A, we have*

$$D_{\mathrm{KL}}(p \parallel p_{\boldsymbol{\theta}}^{\mathrm{SDE}}) \leqslant \mathcal{J}_{\mathrm{SM}}(\boldsymbol{\theta}; g(\cdot)^2) + D_{\mathrm{KL}}(p_T \parallel \pi). \tag{8}$$

*Sketch of proof.* Let $\boldsymbol{\mu}$ and $\boldsymbol{\nu}$ denote the path measures of SDEs in Eq. (1) and Eq. (5) respectively. Intuitively, $\boldsymbol{\mu}$ is the joint distribution of the diffusion process $\{\mathbf{x}(t)\}_{t\in[0,T]}$ given in Section 2.1, and $\boldsymbol{\nu}$ represents the joint distribution of the process $\{\hat{\mathbf{x}}_{\boldsymbol{\theta}}(t)\}_{t\in[0,T]}$ defined in Section 3. Since we can marginalize $\boldsymbol{\mu}$ and $\boldsymbol{\nu}$ to obtain distributions $p$ and $p_{\boldsymbol{\theta}}^{\mathrm{SDE}}$, the data processing inequality gives $D_{\mathrm{KL}}(p \parallel p_{\boldsymbol{\theta}}^{\mathrm{SDE}}) \leqslant D_{\mathrm{KL}}(\boldsymbol{\mu} \parallel \boldsymbol{\nu})$. From the chain rule for the KL divergence, we also have $D_{\mathrm{KL}}(\boldsymbol{\mu} \parallel \boldsymbol{\nu}) = D_{\mathrm{KL}}(p_T \parallel \pi) + \mathbb{E}_{p_T(\mathbf{z})}[D_{\mathrm{KL}}(\boldsymbol{\mu}(\cdot \mid \mathbf{x}(T) = \mathbf{z}) \parallel \boldsymbol{\nu}(\cdot \mid \hat{\mathbf{x}}_{\boldsymbol{\theta}}(T) = \mathbf{z}))]$, where the KL divergence in the final term can be computed by applying the Girsanov theorem [34] to Eq. (5) and the reverse-time SDE of Eq. (1). $\qquad\square$

When the prior distribution $\pi$ is fixed, Theorem 1 guarantees that optimizing the weighted combination of score matching losses $\mathcal{J}_{\mathrm{SM}}(\boldsymbol{\theta}; g(\cdot)^2)$ is equivalent to minimizing an upper bound on the KL divergence from the data distribution $p$ to the model distribution $p_{\boldsymbol{\theta}}^{\mathrm{SDE}}$. Due to well-known equivalence between minimizing KL divergence and maximizing likelihood, we have the following corollary.

**Corollary 1.** *Consider the same conditions and notations in Theorem 1. When $\pi$ is a fixed prior distribution that does not depend on $\boldsymbol{\theta}$, we have*

$$-\mathbb{E}_{p(\mathbf{x})}[\log p_{\boldsymbol{\theta}}^{\mathrm{SDE}}(\mathbf{x})] \leqslant \mathcal{J}_{\mathrm{SM}}(\boldsymbol{\theta}; g(\cdot)^2) + C_1 = \mathcal{J}_{\mathrm{DSM}}(\boldsymbol{\theta}; g(\cdot)^2) + C_2,$$

*where $C_1$ and $C_2$ are constants independent of $\boldsymbol{\theta}$.*

In light of the result in Corollary 1, we henceforth term $\lambda(t) = g(t)^2$ the *likelihood weighting*. The original weighting functions in [48] are inspired from earlier work such as [44, 45] and [19], which are motivated by balancing different score matching losses in the combination, and justified by empirical performance. In contrast, likelihood weighting is motivated from maximizing the likelihood of a probabilistic model induced by the diffusion process, and derived by theoretical analysis. There are three types of SDEs considered in [48]: the Variance Exploding (VE) SDE, the Variance Preserving (VP) SDE, and the subVP SDE. In Table 1, we summarize all these SDEs and contrast their original weighting functions with our likelihood weighting. For VE SDE, our likelihood weighting incidentally coincides with the original weighting used in [48], whereas for VP and subVP SDEs they differ from one another.

Theorem 1 leaves two questions unanswered. First, what are the conditions for the bound to be tight (become an equality)? Second, is there any connection between $p_{\boldsymbol{\theta}}^{\mathrm{SDE}}$ and $p_{\boldsymbol{\theta}}^{\mathrm{ODE}}$ under some conditions? We provide both answers in the following theorem.

**Theorem 2.** *Suppose $p(\mathbf{x})$ and $q(\mathbf{x})$ have continuous second-order derivatives and finite second moments. Let $\{\mathbf{x}(t)\}_{t\in[0,T]}$ be the diffusion process defined by the SDE in Eq. (1). We use $p_t$ and $q_t$ to denote the distributions of $\mathbf{x}(t)$ when $\mathbf{x}(0) \sim p$ and $\mathbf{x}(0) \sim q$, and assume they satisfy the same assumptions in Appendix A. Under the conditions $q_T = \pi$ and $\boldsymbol{s}_{\boldsymbol{\theta}}(\mathbf{x}, t) \equiv \nabla_{\mathbf{x}} \log q_t(\mathbf{x})$ for all $t \in [0, T]$, we have the following equivalence in distributions*

$$p_{\boldsymbol{\theta}}^{\mathrm{SDE}} = p_{\boldsymbol{\theta}}^{\mathrm{ODE}} = q. \tag{9}$$

*Moreover, we have*

$$D_{\mathrm{KL}}(p \parallel p_{\boldsymbol{\theta}}^{\mathrm{SDE}}) = \mathcal{J}_{\mathrm{SM}}(\boldsymbol{\theta}; g(\cdot)^2) + D_{\mathrm{KL}}(p_T \parallel \pi). \tag{10}$$

*Sketch of proof.* When $s_\theta(\mathbf{x}, t)$ matches $\nabla_\mathbf{x} \log q_t(\mathbf{x})$, they both represent the time-dependent score of the same stochastic process so we immediately have $p_\theta^{\mathrm{SDE}} = q$. According to the theory of probability flow ODEs, we also have $p_\theta^{\mathrm{ODE}} = q = p_\theta^{\mathrm{SDE}}$. To prove Eq. (10), we note that $D_{\mathrm{KL}}(p \parallel p_\theta^{\mathrm{SDE}}) = D_{\mathrm{KL}}(p\|q) = D_{\mathrm{KL}}(p_T\|q_T) - \int_0^T \frac{\mathrm{d}}{\mathrm{d}t} D_{\mathrm{KL}}(p_t \parallel q_t)\,\mathrm{d}t = D_{\mathrm{KL}}(p_T\|\pi) - \int_0^T \frac{\mathrm{d}}{\mathrm{d}t} D_{\mathrm{KL}}(p_t \parallel q_t)\,\mathrm{d}t$. We can now complete the proof by simplifying the integrand using the Fokker–Planck equation of $p_t$ and $q_t$ followed by integration by parts. $\qquad\square$

In practice, the conditions of Theorem 2 are hard to satisfy since our score-based model $s_\theta(\mathbf{x}, t)$ will not exactly match the score function $\nabla_\mathbf{x} \log q_t(\mathbf{x})$ of some reverse-time diffusion process with the initial distribution $q_T = \pi$. In other words, our score model may not be a valid time-dependent score function of a stochastic process with an appropriate initial distribution. Therefore, although score matching with likelihood weighting performs approximate maximum likelihood training for $p_\theta^{\mathrm{SDE}}$, we emphasize that it is not theoretically guaranteed to make the likelihood of $p_\theta^{\mathrm{ODE}}$ better. That said, $p_\theta^{\mathrm{ODE}}$ will closely match $p_\theta^{\mathrm{SDE}}$ if our score-based model well-approximates the true score such that $s_\theta(\mathbf{x}, t) \approx \nabla_\mathbf{x} \log p_t(\mathbf{x})$ for all $\mathbf{x}$ and $t \in [0, T]$. Moreover, we empirically observe in our experiments (see Table 2) that training with the likelihood weighting is actually able to consistently improve the likelihood of $p_\theta^{\mathrm{ODE}}$ across multiple datasets, SDEs, and model architectures.

## 4.2 Bounding the log-likelihood on individual datapoints

The bound in Theorem 1 is for the entire distributions of $p$ and $p_\theta^{\mathrm{SDE}}$, but we often seek to bound the log-likelihood for an individual data point $\mathbf{x}$. In addition, $\mathcal{J}_{\mathrm{SM}}(\boldsymbol{\theta}; \lambda(\cdot))$ in the bound is not directly computable due to the unknown quantity $\nabla_\mathbf{x} \log p_t(\mathbf{x})$, and can only be evaluated up to an additive constant through $\mathcal{J}_{\mathrm{DSM}}(\boldsymbol{\theta}; \lambda(\cdot))$ (as we already discussed in Section 2.2). Therefore, the bound in Theorem 1 is only suitable for training purposes. To address these issues, we provide the following bounds for individual data points.

**Theorem 3.** *Let $p_{0t}(\mathbf{x}' \mid \mathbf{x})$ denote the transition distribution from $p_0(\mathbf{x})$ to $p_t(\mathbf{x})$ for the SDE in Eq. (1). With the same notations and conditions in Theorem 1, we have*

$$-\log p_\theta^{\mathrm{SDE}}(\mathbf{x}) \leqslant \mathcal{L}_\theta^{SM}(\mathbf{x}) = \mathcal{L}_\theta^{DSM}(\mathbf{x}), \tag{11}$$

*where $\mathcal{L}_\theta^{SM}(\mathbf{x})$ is defined as*

$$-\mathbb{E}_{p_{0T}(\mathbf{x}'|\mathbf{x})}[\log \pi(\mathbf{x}')] + \frac{1}{2} \int_0^T \mathbb{E}_{p_{0t}(\mathbf{x}'|\mathbf{x})} \left[ 2g(t)^2 \nabla_{\mathbf{x}'} \cdot s_\theta(\mathbf{x}', t) + g(t)^2 \left\| s_\theta(\mathbf{x}', t) \right\|_2^2 - 2\nabla_{\mathbf{x}'} \cdot f(\mathbf{x}', t) \right] \mathrm{d}t,$$

*and $\mathcal{L}_\theta^{DSM}(\mathbf{x})$ is given by*

$$-\mathbb{E}_{p_{0T}(\mathbf{x}'|\mathbf{x})}[\log \pi(\mathbf{x}')] + \frac{1}{2} \int_0^T \mathbb{E}_{p_{0t}(\mathbf{x}'|\mathbf{x})} \left[ g(t)^2 \left\| s_\theta(\mathbf{x}', t) - \nabla_{\mathbf{x}'} \log p_{0t}(\mathbf{x}' \mid \mathbf{x}) \right\|_2^2 \right] \mathrm{d}t$$
$$- \frac{1}{2} \int_0^T \mathbb{E}_{p_{0t}(\mathbf{x}'|\mathbf{x})} \left[ g(t)^2 \left\| \nabla_{\mathbf{x}'} \log p_{0t}(\mathbf{x}' \mid \mathbf{x}) \right\|_2^2 + 2\nabla_{\mathbf{x}'} \cdot f(\mathbf{x}', t) \right] \mathrm{d}t.$$

*Sketch of proof.* For any continuous data distribution $p$, we have $-\mathbb{E}_{p(\mathbf{x})}[\log p_\theta^{\mathrm{SDE}}(\mathbf{x})] = D_{\mathrm{KL}}(p \parallel p_\theta^{\mathrm{SDE}}) + \mathcal{H}(p)$, where $\mathcal{H}(p)$ denotes the differential entropy of $p$. The KL term can be bounded according to Theorem 1, while the differential entropy has an identity similar to Theorem 2 (see Theorem 4 in Appendix A). Combining the bound of $D_{\mathrm{KL}}(p \parallel p_\theta^{\mathrm{SDE}})$ and the identity of $\mathcal{H}(p)$, we obtain a bound on $-\mathbb{E}_{p(\mathbf{x})}[\log p_\theta^{\mathrm{SDE}}(\mathbf{x})]$ that holds for all continuous distribution $p$. Removing the expectation over $p$ on both sides then gives us a bound on $-\log p_\theta^{\mathrm{SDE}}(\mathbf{x})$ for an individual datapoint $\mathbf{x}$. We can simplify this bound to $\mathcal{L}_\theta^{\mathrm{SM}}(\mathbf{x})$ and $\mathcal{L}_\theta^{\mathrm{DSM}}(\mathbf{x})$ with similar techniques to [23] and [56]. $\quad\square$

We provide two equivalent bounds $\mathcal{L}_\theta^{\mathrm{SM}}(\mathbf{x})$ and $\mathcal{L}_\theta^{\mathrm{DSM}}(\mathbf{x})$. The former bears resemblance to score matching while the second resembles denoising score matching. Both admit efficient unbiased estimators when $f(\cdot, t)$ is linear, as the time integrals and expectations in $\mathcal{L}_\theta^{\mathrm{SM}}(\mathbf{x})$ and $\mathcal{L}_\theta^{\mathrm{DSM}}(\mathbf{x})$ can be estimated by samples of the form $(t, \mathbf{x}')$, where $t$ is uniformly sampled over $[0, T]$, and $\mathbf{x}' \sim p_{0t}(\mathbf{x}' \mid \mathbf{x})$. Since the transition distribution $p_{0t}(\mathbf{x}' \mid \mathbf{x})$ is a tractable Gaussian when $f(\cdot, t)$ is linear, we can easily sample from it as well as evaluating $\nabla_{\mathbf{x}'} \log p_{0t}(\mathbf{x}' \mid \mathbf{x})$ for computing $\mathcal{L}_\theta^{\mathrm{DSM}}(\mathbf{x})$.

Moreover, the divergences $\nabla_{\mathbf{x}} \cdot \boldsymbol{s_\theta}(\mathbf{x}, t)$ and $\nabla_{\mathbf{x}} \cdot \boldsymbol{f}(\mathbf{x}, t)$ in $\mathcal{L}_\theta^{\text{SM}}(\mathbf{x})$ and $\mathcal{L}_\theta^{\text{DSM}}(\mathbf{x})$ have efficient unbiased estimators via the Skilling–Hutchinson trick [42, 22].

We can view $\mathcal{L}_\theta^{\text{DSM}}(\mathbf{x})$ as a continuous-time generalization of the evidence lower bound (ELBO) in diffusion probabilistic models [43, 19]. Our bounds in Theorem 3 are not only useful for optimizing and estimating $\log p_\theta^{\text{SDE}}(\mathbf{x})$, but also for training the drift and diffusion coefficients $\boldsymbol{f}(\mathbf{x}, t)$ and $g(t)$ jointly with the score-based model $\boldsymbol{s_\theta}(\mathbf{x}, t)$; we leave this avenue of research for future work. In addition, we can plug the bounds in Theorem 3 into any objective that involves minimizing $-\log p_\theta^{\text{SDE}}(\mathbf{x})$ to obtain an efficient surrogate. Section 5.2 provides an example, where we perform variational dequantization to further improve the likelihood of score-based diffusion models.

Similar to the observation in Section 4.1, $\mathcal{L}_\theta^{\text{SM}}(\mathbf{x})$ and $\mathcal{L}_\theta^{\text{DSM}}(\mathbf{x})$ are not guaranteed to upper bound $-\log p_\theta^{\text{ODE}}(\mathbf{x})$. However, they should become approximate upper bounds when $\boldsymbol{s_\theta}(\mathbf{x}, t)$ is trained sufficiently close to the ground truth. In fact, we empirically observe that $-\log p_\theta^{\text{ODE}}(\mathbf{x}) \leqslant \mathcal{L}_\theta^{\text{SM}}(\mathbf{x}) = \mathcal{L}_\theta^{\text{DSM}}(\mathbf{x})$ holds true for $\mathbf{x}$ sampled from the dataset in all experiments.

### 4.3 Numerical stability

So far we have assumed that the SDEs are defined in the time horizon $[0, T]$ in all theoretical analysis. In practice, however, we often face numerical instabilities when $t \to 0$. To avoid them, we choose a small non-zero starting time $\epsilon > 0$, and train/evaluate score-based diffusion models in the time horizon $[\epsilon, T]$ instead of $[0, T]$. Since $\epsilon$ is small, training score-based diffusion models with likelihood weighting still approximately maximizes their model likelihood. Yet at test time, the likelihood bound as computed in Theorem 3 is slightly biased, rendering the values not directly comparable to results reported in other works. We use Jensen's inequality to correct for this bias in our experiments, for which we provide a detailed explanation in Appendix B.

### 4.4 Related work

Our result in Theorem 2 can be viewed as a generalization of De Bruijin's identity ([49], Eq. 2.12) from its original differential form to an integral form. De Bruijn's identity relates the rate of change of the Shannon entropy under an additive Gaussian noise channel to the Fisher information, a result which can be interpreted geometrically as relating the rate of change of the volume of a distribution's typical set to its surface area. Ref. [2] (Lemma 1) builds on this result and presents an integral and relative form of de Bruijn's identity which relates the KL divergence to the integral of the relative Fisher information for a distribution of interest and a reference standard normal. More generally, various identities and inequalities involving the (relative) Shannon entropy and (relative) Fisher information have found use in proofs of the central limit theorem [24]. Ref. [31] (Theorem 1) covers similar ground to the relative form of de Bruijn's identity, but is perhaps the first to consider its implications for learning in probabilistic models by framing the discussion in terms of the score matching objective ([23], Eq. 2).

## 5 Improving the likelihood of score-based diffusion models

Our theoretical analysis implies that training with the likelihood weighting should improve the likelihood of score-based diffusion models. To verify this empirically, we test likelihood weighting with different model architectures, SDEs, and datasets. We observe that switching to likelihood weighting increases the variance of the training objective and propose to counteract it with importance sampling. We additionally combine our bound with variational dequantization [18] which narrows the gap between the likelihood of continuous and discrete probability models. All combined, we observe consistent improvement of likelihoods for both $p_\theta^{\text{SDE}}$ and $p_\theta^{\text{ODE}}$ across all settings. We term the model $p_\theta^{\text{ODE}}$ trained in this way *ScoreFlow*, and show that it achieves excellent likelihoods on CIFAR-10 [28] and ImageNet $32 \times 32$ [55], on a par with cutting-edge autoregressive models.

### 5.1 Variance reduction via importance sampling

As mentioned in Section 2.2, we typically use Monte Carlo sampling to approximate the time integral in $\mathcal{J}_{\text{DSM}}(\boldsymbol{\theta}; \lambda(\cdot))$ during training. In particular, we first uniformly sample a time step $t \sim \mathcal{U}[0, T]$, and then use the denoising score matching loss at $t$ as an estimate for the whole time integral. This

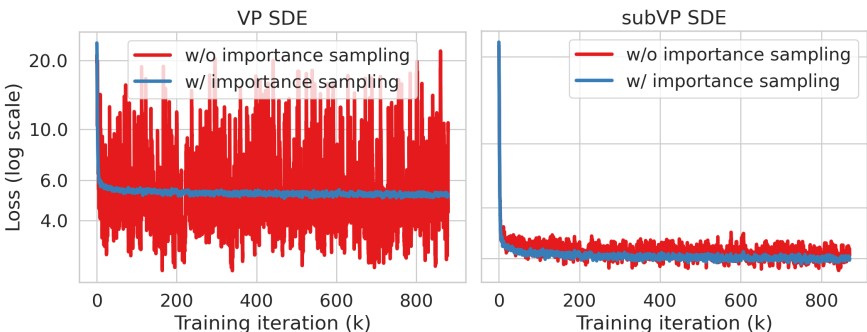

Figure 2: Learning curves with the likelihood weighting on the CIFAR-10 dataset (smoothed with exponential moving average). Importance sampling significantly reduces the loss variance.

Monte Carlo approximation is much faster than computing the time integral accurately, but introduces additional variance to the training loss.

We empirically observe that this Monte Carlo approximation suffers from a larger variance when using our likelihood weighting instead of the original weightings in [48]. Leveraging importance sampling, we propose a new Monte Carlo approximation that significantly reduces the variance of learning curves under likelihood weighting, as demonstrated in Fig. 2. In fact, with importance sampling, the loss variance (after convergence) decreases from 98.48 to 0.068 on CIFAR-10, and decreases from 0.51 to 0.043 on ImageNet.

Let $\lambda(t) = \alpha(t)^2$ denote the weightings in [48] (reproduced in Table 1), and recall that our likelihood weighting is $\lambda(t) = g(t)^2$. Since $\alpha(t)^2$ empirically leads to lower variance, we can use a proposal distribution $p(t) := \frac{g(t)^2}{\alpha(t)^2 Z}$ to change the weighting in $\mathcal{J}_{\text{DSM}}(\boldsymbol{\theta}; g(\cdot)^2)$ from $g(t)^2$ to $\alpha(t)^2$ with importance sampling, where $Z$ is a normalizing constant that ensures $\int p(t)\,\mathrm{d}t = 1$. Specifically, for any function $h(t)$, we estimate the time integral $\int_0^T g(t)^2 h(t)\,\mathrm{d}t$ with

$$\int_0^T g(t)^2 h(t)\,\mathrm{d}t = Z\int_0^T p(t)\alpha(t)^2 h(t)\,\mathrm{d}t \approx TZ\alpha(\tilde{t})^2 h(\tilde{t}), \tag{12}$$

where $\tilde{t}$ is a sample from $p(t)$. When training score-based models with likelihood weighting, $h(t)$ corresponds to the denoising score matching loss at time $t$.

Ref. [33] also observes that optimizing the ELBO for diffusion probabilistic models has large variance, and proposes to reduce it with importance sampling. They build their proposal distribution based on historical loss values stored at thousands of discrete time steps. Despite this similarity, our method is easier to implement without needing to maintain history, can be used for evaluation, and is particularly suited to the continuous-time setting.

## 5.2 Variational dequantization

Digital images are discrete data, and must be dequantized when training continuous density models like normalizing flows [12, 13] and score-based diffusion models. One popular approach to this is uniform dequantization [53, 50], where we add small uniform noise over $[0, 1)$ to images taking values in $\{0, 1, \cdots, 255\}$. As shown in [50], training a continuous model $p_{\boldsymbol{\theta}}(\mathbf{x})$ on uniformly dequantized data implicitly maximizes a lower bound on the log-likelihood of a certain discrete model $P_{\boldsymbol{\theta}}(\mathbf{x})$. Due to the gap between $p_{\boldsymbol{\theta}}(\mathbf{x})$ and $P_{\boldsymbol{\theta}}(\mathbf{x})$, comparing the likelihood of continuous density models to models which fit discrete data directly, such as autoregressive models [55] or variational autoencoders, naturally puts the former at a disadvantage.

To minimize the gap between $p_{\boldsymbol{\theta}}(\mathbf{x})$ and $P_{\boldsymbol{\theta}}(\mathbf{x})$, ref. [18] proposes variational dequantization, where a separate normalizing flow model $q_{\boldsymbol{\phi}}(\mathbf{u} \mid \mathbf{x})$ is trained to produce the dequantization noise by optimizing the following objective

$$\max_{\boldsymbol{\phi}} \mathbb{E}_{\mathbf{x}\sim p(\mathbf{x})}\mathbb{E}_{\mathbf{u}\sim q_{\boldsymbol{\phi}}(\cdot\mid\mathbf{x})}[\log p_{\boldsymbol{\theta}}(\mathbf{x} + \mathbf{u}) - \log q_{\boldsymbol{\phi}}(\mathbf{u} \mid \mathbf{x})]. \tag{13}$$

Plugging in the lower bound on $\log p_{\boldsymbol{\theta}}(\mathbf{x})$ from Theorem 3, we can optimize Eq. (13) to improve the likelihood of score-based diffusion models.

Table 2: Negative log-likelihood (bits/dim) and sample quality (FID scores) on CIFAR-10 and ImageNet $32 \times 32$. Abbreviations: "NLL" for "negative log-likelihood"; "Uni. deq." for "Uniform dequantization"; "Var. deq." for "Variational dequantization"; "LW" for "likelihood weighting"; and "IS" for "importance sampling". Bold indicates best result in the corresponding column. Shaded rows represent models trained with both likelihood weighting and importance sampling.

| Model | SDE | CIFAR-10 | | | | | ImageNet $32 \times 32$ | | | | |
| | | Uni. deq. | | Var. deq. | | FID↓ | Uni. deq. | | Var. deq. | | FID↓ |
| | | NLL↓ | Bound↓ | NLL↓ | Bound↓ | | NLL↓ | Bound↓ | NLL↓ | Bound↓ | |
| Baseline | VP | 3.16 | 3.28 | 3.04 | 3.14 | 3.98 | 3.90 | 3.96 | 3.84 | 3.91 | **8.34** |
| Baseline + LW | VP | 3.06 | 3.18 | 2.94 | 3.03 | 5.18 | 3.91 | 3.96 | 3.86 | 3.92 | 17.75 |
| Baseline + LW + IS | VP | 2.95 | 3.08 | 2.83 | 2.94 | 6.03 | 3.86 | 3.92 | 3.80 | 3.88 | 11.15 |
| Deep | VP | 3.13 | 3.25 | 3.01 | 3.10 | 3.09 | 3.89 | 3.95 | 3.84 | 3.90 | 8.40 |
| Deep + LW | VP | 3.06 | 3.17 | 2.93 | 3.02 | 7.88 | 3.91 | 3.96 | 3.86 | 3.92 | 17.73 |
| Deep + LW + IS | VP | 2.93 | 3.06 | **2.80** | 2.92 | 5.34 | 3.85 | 3.92 | 3.79 | 3.88 | 11.20 |
| Baseline | subVP | 2.99 | 3.09 | 2.88 | 2.98 | 3.20 | 3.87 | 3.92 | 3.82 | 3.88 | 8.71 |
| Baseline + LW | subVP | 2.97 | 3.07 | 2.86 | 2.96 | 7.33 | 3.87 | 3.92 | 3.82 | 3.88 | 12.99 |
| Baseline + LW + IS | subVP | 2.94 | 3.05 | 2.84 | 2.94 | 5.58 | 3.84 | 3.91 | 3.79 | 3.87 | 10.57 |
| Deep | subVP | 2.96 | 3.06 | 2.85 | 2.95 | **2.86** | 3.86 | 3.91 | 3.81 | 3.87 | 8.87 |
| Deep + LW | subVP | 2.95 | 3.05 | 2.85 | 2.94 | 6.57 | 3.88 | 3.93 | 3.83 | 3.88 | 16.55 |
| Deep + LW + IS | subVP | **2.90** | **3.02** | 2.81 | **2.90** | 5.40 | **3.82** | **3.90** | **3.76** | **3.86** | 10.18 |

## 5.3 Experiments

We empirically test the performance of likelihood weighting, importance sampling and variational dequantization across multiple architectures of score-based models, SDEs, and datasets. In particular, we consider DDPM++ ("Baseline" in Table 2) and DDPM++ (deep) ("Deep" in Table 2) models with VP and subVP SDEs [48] on CIFAR-10 [28] and ImageNet $32 \times 32$ [55] datasets. We omit experiments on the VE SDE since (i) under this SDE our likelihood weighting is the same as the original weighting in [48]; (ii) we empirically observe that the best VE SDE model achieves around 3.4 bits/dim on CIFAR-10 in our experiments, which is significantly worse than other SDEs. For each experiment, we report $-\mathbb{E}[\log p_{\boldsymbol{\theta}}^{\text{ODE}}(\mathbf{x})]$ ("Negative log-likelihood" in Table 2), and the upper bound $\mathbb{E}[\mathcal{L}_{\boldsymbol{\theta}}^{\text{DSM}}(\mathbf{x})]$ on $-\mathbb{E}[\log p_{\boldsymbol{\theta}}^{\text{SDE}}(\mathbf{x})]$ ("Bound" in Table 2). In addition, we report FID scores [17] for samples from $p_{\boldsymbol{\theta}}^{\text{ODE}}$, produced by solving the corresponding ODE with the Dormand–Prince RK45 [14] solver. Unless otherwise noted, we apply horizontal flipping as data augmentation for training models on CIFAR-10, so as to match the settings in [48, 19]. Detailed description of all our experiments can be found in Appendices B and C.

We summarize all results in Table 2. Our key observations are as follows:

1. Although Theorem 3 only guarantees $\mathbb{E}[\mathcal{L}_{\boldsymbol{\theta}}^{\text{DSM}}(\mathbf{x})] \geqslant -\mathbb{E}[\log p_{\boldsymbol{\theta}}^{\text{SDE}}(\mathbf{x})]$, and in general we have $p_{\boldsymbol{\theta}}^{\text{SDE}} \neq p_{\boldsymbol{\theta}}^{\text{ODE}}$, we still find that $\mathbb{E}[\mathcal{L}_{\boldsymbol{\theta}}^{\text{DSM}}(\mathbf{x})]$ ("Bound" in Table 2) $\geqslant -\mathbb{E}[\log p_{\boldsymbol{\theta}}^{\text{ODE}}(\mathbf{x})]$ ("NLL" in Table 2) in all our settings.

2. When all conditions are fixed except for the weighting in the training objective, having a lower value of the bound for $p_{\boldsymbol{\theta}}^{\text{SDE}}$ always leads to a lower negative log-likelihood for $p_{\boldsymbol{\theta}}^{\text{ODE}}$.

3. With only likelihood weighting, we can uniformly improve the likelihood of $p_{\boldsymbol{\theta}}^{\text{ODE}}$ and the bound of $p_{\boldsymbol{\theta}}^{\text{SDE}}$ on CIFAR-10 across model architectures and SDEs, but it is not sufficient to guarantee likelihood improvement on ImageNet $32 \times 32$.

4. By combining importance sampling and likelihood weighting, we are able to achieve uniformly better likelihood for $p_{\boldsymbol{\theta}}^{\text{ODE}}$ and bounds for $p_{\boldsymbol{\theta}}^{\text{SDE}}$ across all model architectures, SDEs, and datasets, with only slight degradation of sample quality as measured by FID [17].

5. Variational dequantization uniformly improves both the bound for $p_{\boldsymbol{\theta}}^{\text{SDE}}$ and the negative log-likelihood (NLL) of $p_{\boldsymbol{\theta}}^{\text{ODE}}$ in all settings, regardless of likelihood weighting.

Our experiments confirm that with importance sampling, likelihood weighting is not only effective for maximizing the lower bound for the log-likelihood of $p_{\boldsymbol{\theta}}^{\text{SDE}}$, but also improving the log-likelihood of $p_{\boldsymbol{\theta}}^{\text{ODE}}$. In agreement with [19, 33], we observe that models achieving better likelihood tend to have worse FIDs. However, we emphasize that this degradation of FID is small, and samples actually

have no obvious difference in visual quality (see Figs. 3 and 4). To trade likelihood for FID, we can use weighting functions that interpolate between likelihood weighting and the original weighting functions in [48]. Our FID scores are still much better than most other likelihood-based models.

We term $p_{\boldsymbol{\theta}}^{\text{ODE}}$ a *ScoreFlow* when its corresponding score-based model $s_{\boldsymbol{\theta}}(\mathbf{x}, t)$ is trained with likelihood weighting, importance sampling, and variational dequantization combined. It can be viewed as a continuous normalizing flow, but is parameterized by a score-based model and trained in a more efficient way. With variational dequantization, we show ScoreFlows obtain competitive negative log-likelihoods (NLLs) of 2.83 bits/dim on CIFAR-10 and 3.76 bits/dim on ImageNet $32 \times 32$. Here the ScoreFlow on CIFAR-10 is trained without horizontal flipping (different from the setting in Table 2). As shown in Table 3, our results are on a par with the state-of-the-art autoregressive models on these tasks, and outperform all existing normalizing flow

Table 3: NLLs on CIFAR-10 and ImageNet 32x32.

| Model | CIFAR-10 | ImageNet |
|---|---|---|
| FFJORD [15] | 3.40 | - |
| Flow++ [18] | 3.08 | 3.86 |
| Gated PixelCNN [35] | 3.03 | 3.83 |
| VFlow [4] | 2.98 | 3.83 |
| PixelCNN++ [40] | 2.92 | - |
| NVAE [54] | 2.91 | 3.92 |
| Image Transformer [36] | 2.90 | 3.77 |
| Very Deep VAE [8] | 2.87 | 3.80 |
| PixelSNAIL [7] | 2.85 | 3.80 |
| $\delta$-VAE [38] | 2.83 | 3.77 |
| Sparse Transformer [9] | **2.80** | - |
| ScoreFlow (Ours) | 2.83 | **3.76** |

models. The likelihood for CIFAR-10 can be significantly improved by incorporating advanced data augmentation, as demonstrated in [25, 41]. While we do not compare against these approaches, we believe that incorporating the same data augmentation techniques could also improve the likelihood of ScoreFlows.

## 6 Conclusion

We propose an efficient training objective for approximate maximum likelihood training of score-based diffusion models. Our theoretical analysis shows that the weighted combination of score matching losses upper bounds the negative log-likelihood when using a particular weighting function which we term the likelihood weighting. By minimizing this upper bound, we consistently improve the likelihood of score-based diffusion models across multiple model architectures, SDEs, and datasets. When combined with variational dequantization, we achieve competitive likelihoods on CIFAR-10 and ImageNet $32 \times 32$, matching the performance of best-in-class autoregressive models.

Our upper bound is analogous to the evidence lower bound commonly used for training variational autoencoders. Aside from promoting higher likelihood, the bound can be combined with other objectives that depend on the negative log-likelihood, and also enables joint training of the forward and backward SDEs, which we leave as a future research direction. Our results suggest that score-based diffusion models are competitive alternatives to continuous normalizing flows which enjoy the same tractable likelihood computation but with more efficient maximum likelihood training.

**Limitations and broader impact** Despite promising experimental results, we would like to emphasize that there is no theoretical guarantee that improving the SDE likelihood will improve the ODE likelihood, and this is explicitly a limitation of our work. Score-based diffusion models also suffer from slow sampling. In our experiments, the ODE solver typically need around 550 and 450 evaluations of the score-based model for generation and likelihood computation on CIFAR-10 and ImageNet respectively, which is considerably slower than alternative generative models like VAEs and GANs. In addition, the current formulation of score-based diffusion models only supports continuous data, and cannot be naturally adapted to discrete data without resorting to dequantization. Similarly to other deep generative models, score-based diffusion models can potentially be used to generate harmful media contents such as 'deepfakes', and might reflect and amplify undesirable social bias that could exist in the training dataset.

## Author Contributions

Yang Song wrote the code, ran the experiments, proposed and proved Theorems 1 and 3, and wrote most of the paper. Conor Durkan proposed and proved a first version of Theorem 2, and wrote the paper. Iain Murray and Stefano Ermon co-advised the project and provided helpful edits to the draft.

## Acknowledgments and Disclosure of Funding

The authors would like to thank Sam Power, George Papamakarios, Adji Dieng for helpful feedback, and Duoduo for providing her photos in Fig. 1. This research was supported by NSF (#1651565, #1522054, #1733686), ONR (N000141912145), AFOSR (FA95501910024), ARO (W911NF-21-1-0125), Sloan Fellowship, and Google TPU Research Cloud. This research was also supported by the EPSRC Centre for Doctoral Training in Data Science, funded by the UK Engineering and Physical Sciences Research Council (grant EP/L016427/1), and the University of Edinburgh. Yang Song was supported by the Apple PhD Fellowship in AI/ML.

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
