# A Proofs

We first summarize the notations and assumptions used in our theorems.

**Notations** The drift and diffusion coefficients of the SDE in Eq. (1) are denoted as $\boldsymbol{f} : \mathbb{R}^D \times [0, T] \to \mathbb{R}^D$ and $g : [0, T] \to \mathbb{R}$ respectively, where $[0, T]$ represents a fixed time horizon, and $\times$ denotes the Cartesian product. The solution to Eq. (1) is a stochastic process $\{\mathbf{x}(t)\}_{t \in [0,T]}$. We use $p_t$ to represent the marginal distribution of $\mathbf{x}(t)$, and $p_{0t}(\mathbf{x}' \mid \mathbf{x})$ to denote the transition distribution from $\mathbf{x}(0)$ to $\mathbf{x}(t)$. The data distribution and prior distribution are given by $p$ and $\pi$. We use $\mathcal{C}$ to denote all continuous functions, and let $\mathcal{C}^k$ denote the family of functions with continuous $k$-th order derivatives. For any vector-valued function $\boldsymbol{h} : \mathbb{R}^D \times [0, T] \to \mathbb{R}^D$, we use $\nabla \cdot \boldsymbol{h}(\mathbf{x}, t)$ to represent its divergence with respect to the first input variable.

**Assumptions** We make the following assumptions throughout the paper:

   (i) $p(\mathbf{x}) \in \mathcal{C}^2$ and $\mathbb{E}_{\mathbf{x} \sim p}\big[\|\mathbf{x}\|_2^2\big] < \infty$.

   (ii) $\pi(\mathbf{x}) \in \mathcal{C}^2$ and $\mathbb{E}_{\mathbf{x} \sim \pi}\big[\|\mathbf{x}\|_2^2\big] < \infty$.

   (iii) $\forall t \in [0, T] : \boldsymbol{f}(\cdot, t) \in \mathcal{C}^1, \exists C > 0 \,\forall \mathbf{x} \in \mathbb{R}^D, t \in [0, T] : \|\boldsymbol{f}(\mathbf{x}, t)\|_2 \leqslant C(1 + \|\mathbf{x}\|_2)$.

   (iv) $\exists C > 0, \forall \mathbf{x}, \mathbf{y} \in \mathbb{R}^D : \|\boldsymbol{f}(\mathbf{x}, t) - \boldsymbol{f}(\mathbf{y}, t)\|_2 \leqslant C\|\mathbf{x} - \mathbf{y}\|_2$.

   (v) $g \in \mathcal{C}$ and $\forall t \in [0, T], |g(t)| > 0$.

   (vi) For any open bounded set $\mathcal{O}$, $\int_0^T \int_{\mathcal{O}} \|p_t(\mathbf{x})\|_2^2 + Dg(t)^2 \|\nabla_{\mathbf{x}} p_t(\mathbf{x})\|_2^2 \,\mathrm{d}\mathbf{x}\,\mathrm{d}t < \infty$.

   (vii) $\exists C > 0 \,\forall \mathbf{x} \in \mathbb{R}^D, t \in [0, T] : \|\nabla_{\mathbf{x}} \log p_t(\mathbf{x})\|_2 \leqslant C(1 + \|\mathbf{x}\|_2)$.

  (viii) $\exists C > 0, \forall \mathbf{x}, \mathbf{y} \in \mathbb{R}^D : \|\nabla_{\mathbf{x}} \log p_t(\mathbf{x}) - \nabla_{\mathbf{y}} \log p_t(\mathbf{y})\|_2 \leqslant C\|\mathbf{x} - \mathbf{y}\|_2$.

   (ix) $\exists C > 0 \,\forall \mathbf{x} \in \mathbb{R}^D, t \in [0, T] : \|\boldsymbol{s_\theta}(\mathbf{x}, t)\|_2 \leqslant C(1 + \|\mathbf{x}\|_2)$.

   (x) $\exists C > 0, \forall \mathbf{x}, \mathbf{y} \in \mathbb{R}^D : \|\boldsymbol{s_\theta}(\mathbf{x}, t) - \boldsymbol{s_\theta}(\mathbf{y}, t)\|_2 \leqslant C\|\mathbf{x} - \mathbf{y}\|_2$.

   (xi) Novikov's condition: $\mathbb{E}\Big[\exp\Big(\frac{1}{2}\int_0^T \|\nabla_{\mathbf{x}} \log p_t(\mathbf{x}) - \boldsymbol{s_\theta}(\mathbf{x}, t)\|_2^2 \,\mathrm{d}t\Big)\Big] < \infty$.

  (xii) $\forall t \in [0, T]\,\exists k > 0 : p_t(\mathbf{x}) = O(e^{-\|\mathbf{x}\|_2^k})$ as $\|\mathbf{x}\|_2 \to \infty$.

Below we provide all proofs for our theorems.

**Theorem 1.** *Let $p(\mathbf{x})$ be the data distribution, $\pi(\mathbf{x})$ be a known prior distribution, and $p_{\boldsymbol{\theta}}^{\mathrm{SDE}}$ be defined as in Section 3. Suppose $\{\mathbf{x}(t)\}_{t \in [0,T]}$ is a stochastic process defined by the SDE in Eq. (1) with $\mathbf{x}(0) \sim p$, where the marginal distribution of $\mathbf{x}(t)$ is denoted as $p_t$. Under some regularity conditions detailed in Appendix A, we have*

$$D_{\mathrm{KL}}(p \parallel p_{\boldsymbol{\theta}}^{\mathrm{SDE}}) \leqslant \mathcal{J}_{\mathrm{SM}}(\boldsymbol{\theta}; g(\cdot)^2) + D_{\mathrm{KL}}(p_T \parallel \pi). \tag{8}$$

*Proof.* We denote the path measure of $\{\mathbf{x}(t)\}_{t \in [0,T]}$ and $\{\hat{\mathbf{x}}_{\boldsymbol{\theta}}(t)\}_{t \in [0,T]}$ as $\boldsymbol{\mu}$ and $\boldsymbol{\nu}$ respectively. Due to assumptions (i) (ii) (iii) (iv) (v) (ix) and (x), both $\boldsymbol{\mu}$ and $\boldsymbol{\nu}$ are uniquely given by the corresponding SDEs. Consider a Markov kernel $K(\{\mathbf{z}(t)\}_{t \in [0,t]}, \mathbf{y}) := \delta(\mathbf{z}(0) = \mathbf{y})$. Since $\mathbf{x}(0) \sim p_0$ and $\hat{\mathbf{x}}_{\boldsymbol{\theta}}(0) \sim p_{\boldsymbol{\theta}}$, we have the following result

$$\int K(\{\mathbf{x}(t)\}_{t \in [0,T]}, \mathbf{x})\,\mathrm{d}\boldsymbol{\mu}(\{\mathbf{x}(t)\}_{t \in [0,T]}) = p_0(\mathbf{x})$$

$$\int K(\{\hat{\mathbf{x}}_{\boldsymbol{\theta}}(t)\}_{t \in [0,T]}, \mathbf{x})\,\mathrm{d}\boldsymbol{\nu}(\{\hat{\mathbf{x}}_{\boldsymbol{\theta}}(t)\}_{t \in [0,T]}) = p_{\boldsymbol{\theta}}(\mathbf{x}).$$

Here the Markov kernel $K$ essentially performs marginalization of path measures to obtain "sliced" distributions at $t = 0$. We can use the data processing inequality with this Markov kernel to obtain

$$D_{\mathrm{KL}}(p \parallel p_{\boldsymbol{\theta}}) = D_{\mathrm{KL}}(p_0 \parallel p_{\boldsymbol{\theta}})$$

$$= D_{\mathrm{KL}}\bigg(\int K(\{\mathbf{x}(t)\}_{t \in [0,T]}, \mathbf{x})\,\mathrm{d}\boldsymbol{\mu}(\{\mathbf{x}(t)\}_{t \in [0,T]}) \,\bigg\|\, \int K(\{\hat{\mathbf{x}}_{\boldsymbol{\theta}}(t)\}_{t \in [0,T]}, \mathbf{x})\,\mathrm{d}\boldsymbol{\nu}(\{\hat{\mathbf{x}}_{\boldsymbol{\theta}}(t)\}_{t \in [0,T]})\bigg)$$

$$\leqslant D_{\mathrm{KL}}(\boldsymbol{\mu} \parallel \boldsymbol{\nu}). \tag{14}$$

Recall that by definition $\mathbf{x}(T) \sim p_T$ and $\hat{\mathbf{x}}_{\boldsymbol{\theta}}(T) \sim \pi$. Leveraging the chain rule of KL divergences (see, for example, Theorem 2.4 in [29]), we have

$$D_{\mathrm{KL}}(\boldsymbol{\mu} \parallel \boldsymbol{\nu}) = D_{\mathrm{KL}}(p_T \parallel \pi) + \mathbb{E}_{\mathbf{z} \sim p_T}[D_{\mathrm{KL}}(\boldsymbol{\mu}(\cdot \mid \mathbf{x}(T) = \mathbf{z}) \parallel \boldsymbol{\nu}(\cdot \mid \hat{\mathbf{x}}_{\boldsymbol{\theta}}(T) = \mathbf{z}))]. \qquad (15)$$

Under assumptions (i) (iii) (iv) (v) (vi) (vii) (viii), the SDE in Eq. (1) has a corresponding reverse-time SDE given by

$$\mathrm{d}\mathbf{x} = [\boldsymbol{f}(\mathbf{x}, t) - g(t)^2 \nabla_{\mathbf{x}} \log p_t(\mathbf{x})] \, \mathrm{d}t + g(t) \, \mathrm{d}\bar{\mathbf{w}}. \qquad (16)$$

Since Eq. (16) is the time reversal of Eq. (1), it induces the same path measure $\boldsymbol{\mu}$. As a result, $D_{\mathrm{KL}}(\boldsymbol{\mu}(\cdot \mid \mathbf{x}(T) = \mathbf{z}) \parallel \boldsymbol{\nu}(\cdot \mid \hat{\mathbf{x}}_{\boldsymbol{\theta}}(T) = \mathbf{z}))$ can be viewed as the KL divergence between the path measures induced by the following two (reverse-time) SDEs:

$$\mathrm{d}\mathbf{x} = [\boldsymbol{f}(\mathbf{x}, t) - g(t)^2 \nabla_{\mathbf{x}} \log p_t(\mathbf{x})] \, \mathrm{d}t + g(t) \, \mathrm{d}\bar{\mathbf{w}}, \quad \mathbf{x}(T) = \mathbf{x}$$

$$\mathrm{d}\hat{\mathbf{x}} = [\boldsymbol{f}(\hat{\mathbf{x}}, t) - g(t)^2 \boldsymbol{s}_{\boldsymbol{\theta}}(\hat{\mathbf{x}}, t)] \, \mathrm{d}t + g(t) \, \mathrm{d}\bar{\mathbf{w}}, \quad \hat{\mathbf{x}}_{\boldsymbol{\theta}}(T) = \mathbf{x}.$$

The KL divergence between two SDEs with shared diffusion coefficients and starting points exists under assumptions (vii) (viii) (ix) (x) (xi) (see, *e.g.*, [52, 30]), and can be computed via the Girsanov theorem [34]:

$$D_{\mathrm{KL}}(\boldsymbol{\mu}(\cdot \mid \mathbf{x}(T) = \mathbf{z}) \parallel \boldsymbol{\nu}(\cdot \mid \hat{\mathbf{x}}_{\boldsymbol{\theta}}(T) = \mathbf{z}))$$

$$= -\mathbb{E}_{\boldsymbol{\mu}}\left[ \log \frac{\mathrm{d}\boldsymbol{\nu}}{\mathrm{d}\boldsymbol{\mu}} \right] \qquad (17)$$

$$\overset{(i)}{=} \mathbb{E}_{\boldsymbol{\mu}}\left[ \int_0^T g(t)(\nabla_{\mathbf{x}} \log p_t(\mathbf{x}) - \boldsymbol{s}_{\boldsymbol{\theta}}(\mathbf{x}, t)) \, \mathrm{d}\bar{\mathbf{w}}_t + \frac{1}{2} \int_0^T g(t)^2 \, \|\nabla_{\mathbf{x}} \log p_t(\mathbf{x}) - \boldsymbol{s}_{\boldsymbol{\theta}}(\mathbf{x}, t)\|_2^2 \, \mathrm{d}t \right]$$

$$\overset{(ii)}{=} \mathbb{E}_{\boldsymbol{\mu}}\left[ \frac{1}{2} \int_0^T g(t)^2 \, \|\nabla_{\mathbf{x}} \log p_t(\mathbf{x}) - \boldsymbol{s}_{\boldsymbol{\theta}}(\mathbf{x}, t)\|_2^2 \, \mathrm{d}t \right]$$

$$= \frac{1}{2} \int_0^T \mathbb{E}_{p_t(\mathbf{x})}[g(t)^2 \, \|\nabla_{\mathbf{x}} \log p_t(\mathbf{x}) - \boldsymbol{s}_{\boldsymbol{\theta}}(\mathbf{x}, t)\|_2^2] \, \mathrm{d}t$$

$$= \mathcal{J}_{\mathrm{SM}}(\boldsymbol{\theta}; g(\cdot)^2), \qquad (18)$$

where (i) is due to Girsanov Theorem II [34, Theorem 8.6.6], and (ii) is due to the martingale property of Itô integrals. Combining Eqs. (14), (15) and (18) completes the proof. □

**Theorem 2.** *Suppose $p(\mathbf{x})$ and $q(\mathbf{x})$ have continuous second-order derivatives and finite second moments. Let $\{\mathbf{x}(t)\}_{t \in [0,T]}$ be the diffusion process defined by the SDE in Eq. (1). We use $p_t$ and $q_t$ to denote the distributions of $\mathbf{x}(t)$ when $\mathbf{x}(0) \sim p$ and $\mathbf{x}(0) \sim q$, and assume they satisfy the same assumptions in Appendix A. Under the conditions $q_T = \pi$ and $\boldsymbol{s}_{\boldsymbol{\theta}}(\mathbf{x}, t) \equiv \nabla_{\mathbf{x}} \log q_t(\mathbf{x})$ for all $t \in [0, T]$, we have the following equivalence in distributions*

$$p_{\boldsymbol{\theta}}^{\mathrm{SDE}} = p_{\boldsymbol{\theta}}^{\mathrm{ODE}} = q. \qquad (9)$$

*Moreover, we have*

$$D_{\mathrm{KL}}(p \parallel p_{\boldsymbol{\theta}}^{\mathrm{SDE}}) = \mathcal{J}_{\mathrm{SM}}(\boldsymbol{\theta}; g(\cdot)^2) + D_{\mathrm{KL}}(p_T \parallel \pi). \qquad (10)$$

*Proof.* When $\pi = q_T$ and $\boldsymbol{s}_{\boldsymbol{\theta}}(\mathbf{x}, t) \equiv \nabla_{\mathbf{x}} \log q_t(\mathbf{x})$, the reverse-time SDE that defines $p_{\boldsymbol{\theta}}^{\mathrm{SDE}}$, *i.e.*,

$$\mathrm{d}\hat{\mathbf{x}} = [\boldsymbol{f}(\hat{\mathbf{x}}, t) - g(t)^2 \boldsymbol{s}_{\boldsymbol{\theta}}(\hat{\mathbf{x}}, t)] \, \mathrm{d}t + g(t) \, \mathrm{d}\bar{\mathbf{w}}, \quad \hat{\mathbf{x}}_{\boldsymbol{\theta}}(T) \sim \pi, \qquad (19)$$

becomes equivalent to

$$\mathrm{d}\hat{\mathbf{x}} = [\boldsymbol{f}(\hat{\mathbf{x}}, t) - g(t)^2 \nabla_{\hat{\mathbf{x}}} \log q_t(\hat{\mathbf{x}})] \, \mathrm{d}t + g(t) \, \mathrm{d}\bar{\mathbf{w}}, \quad \hat{\mathbf{x}}_{\boldsymbol{\theta}}(T) \sim q_T, \qquad (20)$$

which yields the same stochastic process as the following forward-time SDE

$$\mathrm{d}\hat{\mathbf{x}} = \boldsymbol{f}(\hat{\mathbf{x}}, t) \, \mathrm{d}t + g(t) \, \mathrm{d}\mathbf{w}, \quad \hat{\mathbf{x}}_{\boldsymbol{\theta}}(0) \sim q. \qquad (21)$$

Since $\hat{\mathbf{x}}_{\boldsymbol{\theta}}(0) \sim p_{\boldsymbol{\theta}}^{\mathrm{SDE}}$ by definition, we immediately have $p_{\boldsymbol{\theta}}^{\mathrm{SDE}} = q$. Similarly, the ODE that defines $p_{\boldsymbol{\theta}}^{\mathrm{ODE}}$ is

$$\frac{\mathrm{d}\tilde{\mathbf{x}}}{\mathrm{d}t} = \boldsymbol{f}_{\boldsymbol{\theta}}(\tilde{\mathbf{x}}, t) - \frac{1}{2} g(t)^2 \boldsymbol{s}_{\boldsymbol{\theta}}(\tilde{\mathbf{x}}, t), \quad \tilde{\mathbf{x}}_{\boldsymbol{\theta}}(T) \sim \pi, \qquad (22)$$

which is equivalent to the following when $q_T = \pi$ and $s_\theta(\mathbf{x}, t) \equiv \nabla_{\mathbf{x}} \log q_t(\mathbf{x})$,

$$\frac{\mathrm{d}\tilde{\mathbf{x}}}{\mathrm{d}t} = \boldsymbol{f}_\theta(\tilde{\mathbf{x}}, t) - \frac{1}{2} g(t)^2 \nabla_{\tilde{\mathbf{x}}} \log q_t(\tilde{\mathbf{x}}, t), \quad \tilde{\mathbf{x}}_\theta(T) \sim q_T. \tag{23}$$

The theory of probability flow ODEs [48] guarantees that Eq. (21) and Eq. (23) share the same set of marginal distributions, $\{q_t\}_{t \in [0,T]}$, which implies that $\tilde{\mathbf{x}}_\theta(0) \sim q$. Since by definition $\tilde{\mathbf{x}}_\theta(0) \sim p_\theta^{\mathrm{ODE}}$, we have $p_\theta^{\mathrm{ODE}} = q$.

The next part of the theorem can be proved by first rewriting the KL divergence from $p$ to $q$ in an integral form:

$$D_{\mathrm{KL}}(p(\mathbf{x}) \| q(\mathbf{x})) \overset{(i)}{=} D_{\mathrm{KL}}(p_0(\mathbf{x}) \| q_0(\mathbf{x})) - D_{\mathrm{KL}}(p_T(\mathbf{x}) \| q_T(\mathbf{x})) + D_{\mathrm{KL}}(p_T(\mathbf{x}) \| q_T(\mathbf{x}))$$

$$\overset{(ii)}{=} \int_T^0 \frac{\partial D_{\mathrm{KL}}(p_t(\mathbf{x}) \| q_t(\mathbf{x}))}{\partial t} \, \mathrm{d}t + D_{\mathrm{KL}}(p_T(\mathbf{x}) \| q_T(\mathbf{x})), \tag{24}$$

where (i) holds due to our definition $p_0(\mathbf{x}) \equiv p(\mathbf{x})$ and $q_0(\mathbf{x}) \equiv q(\mathbf{x})$; (ii) is due to the fundamental theorem of calculus.

Next, we show how to rewrite Eq. (24) as a mixture of score matching losses. The Fokker–Planck equation for the SDE in Eq. (1) describes the time-evolution of the stochastic process's associated probability density function, and is given by

$$\frac{\partial p_t(\mathbf{x})}{\partial t} = \nabla_{\mathbf{x}} \cdot \left( \frac{1}{2} g^2(t) p_t(\mathbf{x}) \nabla_{\mathbf{x}} \log p_t(\mathbf{x}) - \boldsymbol{f}(\mathbf{x}, t) p_t(\mathbf{x}) \right) = \nabla_{\mathbf{x}} \cdot (\boldsymbol{h}_p(\mathbf{x}, t) p_t(\mathbf{x})),$$

where for simplified notations we define $\boldsymbol{h}_p(\mathbf{x}, t) := \frac{1}{2} g^2(t) \nabla_{\mathbf{x}} \log p_t(\mathbf{x}) - \boldsymbol{f}(\mathbf{x}, t)$. Similarly, $\frac{\partial q_t(\mathbf{x})}{\partial t} = \nabla_{\mathbf{x}} \cdot (\boldsymbol{h}_q(\mathbf{x}, t) q_t(\mathbf{x}))$. Since we assume $\log p_t(\mathbf{x})$ and $\log q_t(\mathbf{x})$ are smooth functions with at most polynomial growth at infinity (assumption (xii)), we have $\lim_{\mathbf{x} \to \infty} \boldsymbol{h}_p(\mathbf{x}, t) p_t(\mathbf{x}) = \mathbf{0}$ and $\lim_{\mathbf{x} \to \infty} \boldsymbol{h}_q(\mathbf{x}, t) q_t(\mathbf{x}) = \mathbf{0}$ for all $t$. Then, the time-derivative of $D_{\mathrm{KL}}(p_t \| q_t)$ can be rewritten in the following way:

$$\frac{\partial D_{\mathrm{KL}}(p_t(\mathbf{x}) \| q_t(\mathbf{x}))}{\partial t} = \frac{\partial}{\partial t} \int p_t(\mathbf{x}) \log \frac{p_t(\mathbf{x})}{q_t(\mathbf{x})} \, \mathrm{d}\mathbf{x}$$

$$= \int \frac{\partial p_t(\mathbf{x})}{\partial t} \log \frac{p_t(\mathbf{x})}{q_t(\mathbf{x})} \, \mathrm{d}\mathbf{x} + \underbrace{\int \frac{\partial p_t(\mathbf{x})}{\partial t} \, \mathrm{d}\mathbf{x}}_{=0} - \int \frac{p_t(\mathbf{x})}{q_t(\mathbf{x})} \frac{\partial q_t(\mathbf{x})}{\partial t} \, \mathrm{d}\mathbf{x}$$

$$= \int \nabla_{\mathbf{x}} \cdot (\boldsymbol{h}_p(\mathbf{x}, t) p_t(\mathbf{x})) \log \frac{p_t(\mathbf{x})}{q_t(\mathbf{x})} \, \mathrm{d}\mathbf{x} - \int \frac{p_t(\mathbf{x})}{q_t(\mathbf{x})} \nabla_{\mathbf{x}} \cdot (\boldsymbol{h}_q(\mathbf{x}, t) q_t(\mathbf{x})) \, \mathrm{d}\mathbf{x}$$

$$\overset{(i)}{=} - \int p_t(\mathbf{x}) [\boldsymbol{h}_p^{\mathsf{T}}(\mathbf{x}, t) - \boldsymbol{h}_q^{\mathsf{T}}(\mathbf{x}, t)] [\nabla_{\mathbf{x}} \log p_t(\mathbf{x}) - \nabla_{\mathbf{x}} \log q_t(\mathbf{x})] \, \mathrm{d}\mathbf{x}$$

$$= - \frac{1}{2} \int p_t(\mathbf{x}) g(t)^2 \| \nabla_{\mathbf{x}} \log p_t(\mathbf{x}) - \nabla_{\mathbf{x}} \log q_t(\mathbf{x}) \|_2^2 \, \mathrm{d}\mathbf{x},$$

where (i) is due to integration by parts. Combining with Eq. (24), we can conclude that

$$D_{\mathrm{KL}}(p \| q) = \frac{1}{2} \int_0^T \mathbb{E}_{\mathbf{x} \sim p_t(\mathbf{x})} [g(t)^2 \| \nabla_{\mathbf{x}} \log p_t(\mathbf{x}) - \nabla_{\mathbf{x}} \log q_t(\mathbf{x}) \|_2^2] \, \mathrm{d}t + D_{\mathrm{KL}}(p_T \| q_T). \tag{25}$$

Since $p_\theta^{\mathrm{SDE}} = q$ and $q_T = \pi$, we also have

$$D_{\mathrm{KL}}(p \| p_\theta^{\mathrm{SDE}}) = \frac{1}{2} \int_0^T \mathbb{E}_{\mathbf{x} \sim p_t(\mathbf{x})} [g(t)^2 \| \nabla_{\mathbf{x}} \log p_t(\mathbf{x}) - \nabla_{\mathbf{x}} \log q_t(\mathbf{x}) \|_2^2] \, \mathrm{d}t + D_{\mathrm{KL}}(p_T \| q_T)$$

$$= \mathcal{J}_{\mathrm{SM}}(\theta; g(\cdot)^2) + D_{\mathrm{KL}}(p_T \| q_T), \tag{26}$$

which completes the proof. □

Using a similar technique to Theorem 2, we can express the entropy of a distribution in terms of a time-dependent score function, as detailed in the following theorem.

**Theorem 4.** *Let $\mathcal{H}(p(\mathbf{x}))$ be the differential entropy of the initial probability density $p(\mathbf{x})$. Under the same conditions in Theorem 2, we have*

$$\mathcal{H}(p(\mathbf{x})) = \mathcal{H}(p_T(\mathbf{x})) + \frac{1}{2}\int_0^T \mathbb{E}_{\mathbf{x}\sim p_t(\mathbf{x})}\Big[2\boldsymbol{f}(\mathbf{x},t)^\mathsf{T}\nabla_\mathbf{x}\log p_t(\mathbf{x}) - g(t)^2\left\|\nabla_\mathbf{x}\log p_t(\mathbf{x})\right\|_2^2\Big]\,\mathrm{d}t. \tag{27}$$

$$= \mathcal{H}(p_T(\mathbf{x})) - \frac{1}{2}\int_0^T \mathbb{E}_{\mathbf{x}\sim p_t(\mathbf{x})}\Big[2\nabla\cdot\boldsymbol{f}(\mathbf{x},t) + g(t)^2\left\|\nabla_\mathbf{x}\log p_t(\mathbf{x})\right\|_2^2\Big]\,\mathrm{d}t. \tag{28}$$

*Proof.* Once more we proceed analogously to the proofs of Theorem 2. We have

$$\mathcal{H}(p(\mathbf{x})) - \mathcal{H}(p_T(\mathbf{x})) = \int_T^0 \frac{\partial}{\partial t}\mathcal{H}(p_t(\mathbf{x}))\,\mathrm{d}t. \tag{29}$$

Expanding the integrand, we have

$$
\begin{aligned}
\frac{\partial}{\partial t}\mathcal{H}(p_t(\mathbf{x})) &= -\frac{\partial}{\partial t}\int p_t(\mathbf{x})\log p_t(\mathbf{x})\,\mathrm{d}\mathbf{x} \\
&= -\int \frac{\partial p_t(\mathbf{x})}{\partial t}\log p_t(\mathbf{x}) + \frac{\partial p_t(\mathbf{x})}{\partial t}\,\mathrm{d}\mathbf{x} \\
&= -\int \frac{\partial p_t(\mathbf{x})}{\partial t}\log p_t(\mathbf{x})\,\mathrm{d}\mathbf{x} - \frac{\partial}{\partial t}\underbrace{\int p_t(\mathbf{x})\,\mathrm{d}\mathbf{x}}_{=1} \\
&= -\int \nabla_\mathbf{x}\cdot(\boldsymbol{h}_p(\mathbf{x},t)p_t(\mathbf{x}))\log p_t(\mathbf{x})\,\mathrm{d}\mathbf{x} \\
&\stackrel{(i)}{=} \int p_t(\mathbf{x})\boldsymbol{h}_p^\mathsf{T}(\mathbf{x},t)\nabla_\mathbf{x}\log p_t(\mathbf{x})\,\mathrm{d}\mathbf{x} \\
&= \frac{1}{2}\mathbb{E}_{\mathbf{x}\sim p_t(\mathbf{x})}[g(t)^2\left\|\nabla_\mathbf{x}\log p_t(\mathbf{x})\right\|_2^2 - 2\boldsymbol{f}(\mathbf{x},t)^\mathsf{T}\nabla_\mathbf{x}\log p_t(\mathbf{x})],
\end{aligned}
$$

where again (i) follows from integration by parts and the limiting behaviour of $\boldsymbol{h}_p$ given by assumption (xii). Plugging this expression in for the integrand in Eq. (29) then completes the proof for Eq. (27). For Eq. (28), we can once again perform integration by parts and leverage the limiting behavior of $p_t(\mathbf{x})$ in assumption (xii) to get

$$\mathbb{E}_{p_t(\mathbf{x})}[\boldsymbol{f}(\mathbf{x},t)^\mathsf{T}\nabla_\mathbf{x}\log p_t(\mathbf{x})] = \int \boldsymbol{f}(\mathbf{x},t)^\mathsf{T}\nabla_\mathbf{x}p_t(\mathbf{x})\,\mathrm{d}\mathbf{x} = -\int p_t(\mathbf{x})\nabla\cdot\boldsymbol{f}(\mathbf{x},t)\,\mathrm{d}\mathbf{x},$$

which establishes the equivalence between Eq. (28) and Eq. (27). $\qquad\square$

**Remark** The formula in Theorem 4 provides a new way to estimate the entropy of a data distribution from i.i.d. samples. Specifically, given $\{\mathbf{x}_1, \mathbf{x}_2, \cdots, \mathbf{x}_N\} \stackrel{\text{i.i.d.}}{\sim} p(\mathbf{x})$ and an SDE like Eq. (1), we can first apply score matching to train a time-dependent score-based model such that $\boldsymbol{s}_{\boldsymbol{\theta}}(\mathbf{x},t) \approx \nabla_\mathbf{x}\log p_t(\mathbf{x})$, and then plug $\boldsymbol{s}_{\boldsymbol{\theta}}(\mathbf{x},t)$ into Eq. (27) to obtain the following estimator of $\mathcal{H}(p(\mathbf{x}))$:

$$\mathcal{H}(p_T(\mathbf{x})) + \frac{1}{2N}\sum_{i=1}^N \int_0^T \Big[2\boldsymbol{f}(\mathbf{x}_i,t)^\mathsf{T}\boldsymbol{s}_{\boldsymbol{\theta}}(\mathbf{x}_i,t) - g(t)^2\left\|\boldsymbol{s}_{\boldsymbol{\theta}}(\mathbf{x}_i,t)\right\|_2^2\Big]\,\mathrm{d}t,$$

or plug it into Eq. (28) to obtain the following alternative estimator

$$\mathcal{H}(p_T(\mathbf{x})) - \frac{1}{2N}\sum_{i=1}^N \int_0^T \Big[2\nabla\cdot\boldsymbol{f}(\mathbf{x}_i,t) + g(t)^2\left\|\boldsymbol{s}_{\boldsymbol{\theta}}(\mathbf{x}_i,t)\right\|_2^2\Big]\,\mathrm{d}t.$$

Both estimators can be computed from a score-based model alone, and do not require training a density model.

**Theorem 5.** *Let $p_{0t}(\mathbf{x}' \mid \mathbf{x})$ denote the transition kernel from $p_0(\mathbf{x})$ to $p_t(\mathbf{x})$ for any $t \in (0, T]$. With the same conditions and notations in Theorem 1, we have*

$$-\mathbb{E}_{p(\mathbf{x})}[\log p_{\boldsymbol{\theta}}^{\text{SDE}}(\mathbf{x})] \leqslant -\mathbb{E}_{p_T(\mathbf{x})}[\log \pi(\mathbf{x})] + \frac{1}{2}\int_0^T \mathbb{E}_{\mathbf{x} \sim p_t(\mathbf{x})}[2g(t)^2 \nabla \cdot \boldsymbol{s}_{\boldsymbol{\theta}}(\mathbf{x}, t)$$
$$+ g(t)^2 \left\| \boldsymbol{s}_{\boldsymbol{\theta}}(\mathbf{x}, t) \right\|_2^2 - 2\nabla \cdot \boldsymbol{f}(\mathbf{x}, t)] \, \mathrm{d}t. \tag{30}$$
$$= -\mathbb{E}_{p_T(\mathbf{x})}[\log \pi(\mathbf{x})]$$
$$+ \frac{1}{2}\int_0^T \mathbb{E}_{p_{0t}(\mathbf{x}'|\mathbf{x})p(\mathbf{x})}[g(t)^2 \left\| \boldsymbol{s}_{\boldsymbol{\theta}}(\mathbf{x}', t) - \nabla_{\mathbf{x}'} \log p_{0t}(\mathbf{x}' \mid \mathbf{x}) \right\|_2^2$$
$$- g(t)^2 \left\| \nabla_{\mathbf{x}'} \log p_{0t}(\mathbf{x}' \mid \mathbf{x}) \right\|_2^2 - 2\nabla \cdot \boldsymbol{f}(\mathbf{x}', t)] \, \mathrm{d}t. \tag{31}$$

*Proof.* Since $-\mathbb{E}_{p(\mathbf{x})}[\log p_{\boldsymbol{\theta}}^{\text{SDE}}(\mathbf{x})] = D_{\text{KL}}(p \parallel p_{\boldsymbol{\theta}}^{\text{SDE}}) + \mathcal{H}(p)$, we can combine Theorem 1 and Theorem 4 to obtain

$$-\mathbb{E}_{p(\mathbf{x})}[\log p_{\boldsymbol{\theta}}^{\text{SDE}}(\mathbf{x})] \leqslant \frac{1}{2}\int_0^T \mathbb{E}_{p_t(\mathbf{x})}[g(t)^2 \left\| \nabla_{\mathbf{x}} \log p_t(\mathbf{x}) - \boldsymbol{s}_{\boldsymbol{\theta}}(\mathbf{x}, t) \right\|_2^2] \, \mathrm{d}t + D_{\text{KL}}(p_T \parallel \pi)$$
$$+ \mathcal{H}(p_T(\mathbf{x})) - \frac{1}{2}\int_0^T \mathbb{E}_{p_t(\mathbf{x})}[2\nabla \cdot \boldsymbol{f}(\mathbf{x}, t) + g(t)^2 \left\| \nabla_{\mathbf{x}} \log p_t(\mathbf{x}) \right\|_2^2] \, \mathrm{d}t$$
$$= -\mathbb{E}_{p_T(\mathbf{x})}[\log \pi(\mathbf{x})]$$
$$+ \frac{1}{2}\int_0^T \mathbb{E}_{p_t(\mathbf{x})}[g(t)^2 \left\| \nabla_{\mathbf{x}} \log p_t(\mathbf{x}) - \boldsymbol{s}_{\boldsymbol{\theta}}(\mathbf{x}, t) \right\|_2^2 - g(t)^2 \left\| \nabla_{\mathbf{x}} \log p_t(\mathbf{x}) \right\|_2^2] \, \mathrm{d}t$$
$$- \int_0^T \mathbb{E}_{p_t(\mathbf{x})}[\nabla \cdot \boldsymbol{f}(\mathbf{x}, t)] \, \mathrm{d}t. \tag{32}$$

The second term of Eq. (32) can be simplified via integration by parts

$$\frac{1}{2}\int_0^T \mathbb{E}_{p_t(\mathbf{x})}[g(t)^2 \left\| \nabla_{\mathbf{x}} \log p_t(\mathbf{x}) - \boldsymbol{s}_{\boldsymbol{\theta}}(\mathbf{x}, t) \right\|_2^2 - g(t)^2 \left\| \nabla_{\mathbf{x}} \log p_t(\mathbf{x}) \right\|_2^2] \, \mathrm{d}t$$
$$= \frac{1}{2}\int_0^T \mathbb{E}_{p_t(\mathbf{x})}[g(t)^2 \left\| \boldsymbol{s}_{\boldsymbol{\theta}}(\mathbf{x}, t) \right\|_2^2 - 2g(t)^2 \boldsymbol{s}_{\boldsymbol{\theta}}(\mathbf{x}, t)^{\mathsf{T}} \nabla_{\mathbf{x}} \log p_t(\mathbf{x})] \, \mathrm{d}t$$
$$= \frac{1}{2}\int_0^T \mathbb{E}_{p_t(\mathbf{x})}[g(t)^2 \left\| \boldsymbol{s}_{\boldsymbol{\theta}}(\mathbf{x}, t) \right\|_2^2] \, \mathrm{d}t - \int_0^T \mathbb{E}_{p_t(\mathbf{x})}[g(t)^2 \boldsymbol{s}_{\boldsymbol{\theta}}(\mathbf{x}, t)^{\mathsf{T}} \nabla_{\mathbf{x}} \log p_t(\mathbf{x})] \, \mathrm{d}t$$
$$= \frac{1}{2}\int_0^T \mathbb{E}_{p_t(\mathbf{x})}[g(t)^2 \left\| \boldsymbol{s}_{\boldsymbol{\theta}}(\mathbf{x}, t) \right\|_2^2] \, \mathrm{d}t - \int_0^T g(t)^2 \int p_t(\mathbf{x}) \boldsymbol{s}_{\boldsymbol{\theta}}(\mathbf{x}, t)^{\mathsf{T}} \nabla_{\mathbf{x}} \log p_t(\mathbf{x}) \, \mathrm{d}\mathbf{x} \, \mathrm{d}t$$
$$= \frac{1}{2}\int_0^T \mathbb{E}_{p_t(\mathbf{x})}[g(t)^2 \left\| \boldsymbol{s}_{\boldsymbol{\theta}}(\mathbf{x}, t) \right\|_2^2] \, \mathrm{d}t - \int_0^T g(t)^2 \int \boldsymbol{s}_{\boldsymbol{\theta}}(\mathbf{x}, t)^{\mathsf{T}} \nabla_{\mathbf{x}} p_t(\mathbf{x}) \, \mathrm{d}\mathbf{x} \, \mathrm{d}t$$
$$\overset{(i)}{=} \frac{1}{2}\int_0^T \mathbb{E}_{p_t(\mathbf{x})}[g(t)^2 \left\| \boldsymbol{s}_{\boldsymbol{\theta}}(\mathbf{x}, t) \right\|_2^2] \, \mathrm{d}t + \int_0^T g(t)^2 \int p_t(\mathbf{x}) \nabla \cdot \boldsymbol{s}_{\boldsymbol{\theta}}(\mathbf{x}, t) \, \mathrm{d}\mathbf{x} \, \mathrm{d}t$$
$$= \frac{1}{2}\int_0^T \mathbb{E}_{p_t(\mathbf{x})}[g(t)^2 \left\| \boldsymbol{s}_{\boldsymbol{\theta}}(\mathbf{x}, t) \right\|_2^2 + 2g(t)^2 \nabla \cdot \boldsymbol{s}_{\boldsymbol{\theta}}(\mathbf{x}, t)] \, \mathrm{d}t, \tag{33}$$

where (i) is due to integration by parts and the limiting behavior of $p_t(\mathbf{x})$ given by assumption (xii). Combining Eq. (33) and Eq. (32) completes the proof for Eq. (30).

The proof for Eq. (31) parallels that of denoising score matching [56]. Observe that $p_t(\mathbf{x}) = \int p(\mathbf{x}')p_{0t}(\mathbf{x} \mid \mathbf{x}')\,\mathrm{d}\mathbf{x}'$. As a result,

$$\int_0^T \mathbb{E}_{p_t(\mathbf{x})}[g(t)^2 \boldsymbol{s_\theta}(\mathbf{x},t)^\mathsf{T} \nabla_\mathbf{x} \log p_t(\mathbf{x})]\,\mathrm{d}t = \int_0^T g(t)^2 \int \boldsymbol{s_\theta}(\mathbf{x},t)^\mathsf{T} \nabla_\mathbf{x} p_t(\mathbf{x})\,\mathrm{d}\mathbf{x}\,\mathrm{d}t$$

$$= \int_0^T g(t)^2 \int \boldsymbol{s_\theta}(\mathbf{x},t)^\mathsf{T} \nabla_\mathbf{x} \int p(\mathbf{x}')p_{0t}(\mathbf{x} \mid \mathbf{x}')\,\mathrm{d}\mathbf{x}'\,\mathrm{d}\mathbf{x}\,\mathrm{d}t$$

$$= \int_0^T g(t)^2 \int \boldsymbol{s_\theta}(\mathbf{x},t)^\mathsf{T} \int p(\mathbf{x}') \nabla_\mathbf{x} p_{0t}(\mathbf{x} \mid \mathbf{x}')\,\mathrm{d}\mathbf{x}'\,\mathrm{d}\mathbf{x}\,\mathrm{d}t$$

$$= \int_0^T g(t)^2 \int \boldsymbol{s_\theta}(\mathbf{x},t)^\mathsf{T} \int p(\mathbf{x}') p_{0t}(\mathbf{x} \mid \mathbf{x}') \nabla_\mathbf{x} \log p_{0t}(\mathbf{x} \mid \mathbf{x}')\,\mathrm{d}\mathbf{x}'\,\mathrm{d}\mathbf{x}\,\mathrm{d}t$$

$$= \int_0^T \mathbb{E}_{p(\mathbf{x})p_{0t}(\mathbf{x}'|\mathbf{x})}[g(t)^2 \boldsymbol{s_\theta}(\mathbf{x}',t)^\mathsf{T} \nabla_{\mathbf{x}'} \log p_{0t}(\mathbf{x}' \mid \mathbf{x})]\,\mathrm{d}t. \tag{34}$$

Substituting Eq. (34) into the second term of Eq. (32), we have

$$\frac{1}{2}\int_0^T \mathbb{E}_{p_t(\mathbf{x})}[g(t)^2 \left\| \nabla_\mathbf{x} \log p_t(\mathbf{x}) - \boldsymbol{s_\theta}(\mathbf{x},t) \right\|_2^2 - g(t)^2 \left\| \nabla_\mathbf{x} \log p_t(\mathbf{x}) \right\|_2^2]\,\mathrm{d}t$$

$$= \frac{1}{2}\int_0^T \mathbb{E}_{p_t(\mathbf{x})}[g(t)^2 \left\| \boldsymbol{s_\theta}(\mathbf{x},t) \right\|_2^2 - 2g(t)^2 \boldsymbol{s_\theta}(\mathbf{x},t)^\mathsf{T} \nabla_\mathbf{x} \log p_t(\mathbf{x})]\,\mathrm{d}t$$

$$= \frac{1}{2}\int_0^T \mathbb{E}_{p_t(\mathbf{x})}[g(t)^2 \left\| \boldsymbol{s_\theta}(\mathbf{x},t) \right\|_2^2 - 2g(t)^2 \boldsymbol{s_\theta}(\mathbf{x},t)^\mathsf{T} \nabla_\mathbf{x} \log p_t(\mathbf{x})]\,\mathrm{d}t$$

$$= \frac{1}{2}\int_0^T \mathbb{E}_{p(\mathbf{x})p_{0t}(\mathbf{x}'|\mathbf{x})}[g(t)^2 \left\| \boldsymbol{s_\theta}(\mathbf{x}',t) \right\|_2^2 - 2g(t)^2 \boldsymbol{s_\theta}(\mathbf{x}',t)^\mathsf{T} \nabla_{\mathbf{x}'} \log p_{0t}(\mathbf{x}' \mid \mathbf{x})]\,\mathrm{d}t$$

$$= \frac{1}{2}\int_0^T \mathbb{E}_{p(\mathbf{x})p_{0t}(\mathbf{x}'|\mathbf{x})}[g(t)^2 \left\| \boldsymbol{s_\theta}(\mathbf{x}',t) - \nabla_{\mathbf{x}'} \log p_{0t}(\mathbf{x}' \mid \mathbf{x}) \right\|_2^2 - g(t)^2 \left\| \nabla_{\mathbf{x}'} \log p_{0t}(\mathbf{x}' \mid \mathbf{x}) \right\|_2^2]\,\mathrm{d}t. \tag{35}$$

We can now complete the proof for Eq. (31) by combining Eq. (35) an Eq. (32). □

**Theorem 3.** *Let $p_{0t}(\mathbf{x}' \mid \mathbf{x})$ denote the transition distribution from $p_0(\mathbf{x})$ to $p_t(\mathbf{x})$ for the SDE in Eq. (1). With the same notations and conditions in Theorem 1, we have*

$$- \log p_{\boldsymbol{\theta}}^{\mathrm{SDE}}(\mathbf{x}) \leqslant \mathcal{L}_{\boldsymbol{\theta}}^{SM}(\mathbf{x}) = \mathcal{L}_{\boldsymbol{\theta}}^{DSM}(\mathbf{x}), \tag{11}$$

*where $\mathcal{L}_{\boldsymbol{\theta}}^{SM}(\mathbf{x})$ is defined as*

$$-\mathbb{E}_{p_{0T}(\mathbf{x}'|\mathbf{x})}[\log \pi(\mathbf{x}')] + \frac{1}{2}\int_0^T \mathbb{E}_{p_{0t}(\mathbf{x}'|\mathbf{x})}\left[ 2g(t)^2 \nabla_{\mathbf{x}'} \cdot \boldsymbol{s_\theta}(\mathbf{x}',t) + g(t)^2 \left\| \boldsymbol{s_\theta}(\mathbf{x}',t) \right\|_2^2 - 2\nabla_{\mathbf{x}'} \cdot \boldsymbol{f}(\mathbf{x}',t) \right]\mathrm{d}t,$$

*and $\mathcal{L}_{\boldsymbol{\theta}}^{DSM}(\mathbf{x})$ is given by*

$$- \mathbb{E}_{p_{0T}(\mathbf{x}'|\mathbf{x})}[\log \pi(\mathbf{x}')] + \frac{1}{2}\int_0^T \mathbb{E}_{p_{0t}(\mathbf{x}'|\mathbf{x})}\left[ g(t)^2 \left\| \boldsymbol{s_\theta}(\mathbf{x}',t) - \nabla_{\mathbf{x}'} \log p_{0t}(\mathbf{x}' \mid \mathbf{x}) \right\|_2^2 \right]\mathrm{d}t$$

$$- \frac{1}{2}\int_0^T \mathbb{E}_{p_{0t}(\mathbf{x}'|\mathbf{x})}\left[ g(t)^2 \left\| \nabla_{\mathbf{x}'} \log p_{0t}(\mathbf{x}' \mid \mathbf{x}) \right\|_2^2 + 2\nabla_{\mathbf{x}'} \cdot \boldsymbol{f}(\mathbf{x}',t) \right]\mathrm{d}t.$$

*Proof.* The result in Theorem 5 can be re-written as

$$-\mathbb{E}_{p(\mathbf{x})}[\log p_{\boldsymbol{\theta}}^{\mathrm{SDE}}(\mathbf{x})] \leqslant - \mathbb{E}_{p(\mathbf{x})p_{0T}(\mathbf{x}'|\mathbf{x})}[\log \pi(\mathbf{x}')] + \frac{1}{2}\int_0^T \mathbb{E}_{p(\mathbf{x})p_{0t}(\mathbf{x}'|\mathbf{x})}[2g(t)^2 \nabla \cdot \boldsymbol{s_\theta}(\mathbf{x}',t)$$

$$+ g(t)^2 \left\| \boldsymbol{s_\theta}(\mathbf{x}',t) \right\|_2^2 - 2\nabla \cdot \boldsymbol{f}(\mathbf{x}',t)]\,\mathrm{d}t.$$

$$= - \mathbb{E}_{p(\mathbf{x})p_{0T}(\mathbf{x}'|\mathbf{x})}[\log \pi(\mathbf{x}')]$$

$$+ \frac{1}{2}\int_0^T \mathbb{E}_{p(\mathbf{x})p_{0t}(\mathbf{x}'|\mathbf{x})}[g(t)^2 \left\| \boldsymbol{s_\theta}(\mathbf{x}',t) - \nabla_{\mathbf{x}'} \log p_{0t}(\mathbf{x}' \mid \mathbf{x}) \right\|_2^2$$

$$- g(t)^2 \left\| \nabla_{\mathbf{x}'} \log p_{0t}(\mathbf{x}' \mid \mathbf{x}) \right\|_2^2 - 2\nabla \cdot \boldsymbol{f}(\mathbf{x}',t)]\,\mathrm{d}t.$$

Given a fixed SDE (and its transition kernel $p_{0t}(\mathbf{x}' \mid \mathbf{x})$), Theorem 5 holds for any data distribution $p$ that satisfies our assumptions. Leveraging proof by contradiction, we can easily see that $\mathbb{E}_{p(\mathbf{x})}$ in both sides of Eqs. (30) and (31) can be cancelled to get

$$-\log p_{\boldsymbol{\theta}}^{\text{SDE}}(\mathbf{x}) \leqslant \mathcal{L}_{\boldsymbol{\theta}}^{\text{SM}}(\mathbf{x}) = \mathcal{L}_{\boldsymbol{\theta}}^{\text{DSM}}(\mathbf{x}),$$

which finishes the proof. $\qquad\square$

## B  Numerical stability

In our previous theoretical discussion, we always assume that data are perturbed with an SDE starting from $t = 0$. However, in practical implementations, $t = 0$ often leads to numerical instability. As a pragmatic solution, we choose a small non-zero starting time $\epsilon > 0$, and consider the SDE in the time horizon $[\epsilon, T]$. Using the same proof techniques, we can easily see that when the time horizon is $[\epsilon, T]$ instead of $[0, T]$, the original bound in Theorem 1,

$$D_{\text{KL}}(p \parallel p_{\boldsymbol{\theta}}^{\text{SDE}}) \leqslant \mathcal{J}_{\text{SM}}(\boldsymbol{\theta}; g(\cdot)^2) + D_{\text{KL}}(p_T \parallel \pi)$$
$$= \frac{1}{2} \int_0^T \mathbb{E}_{p_t(\mathbf{x})}[g(t)^2 \|\nabla_{\mathbf{x}} \log p_t(\mathbf{x}) - \boldsymbol{s}_{\boldsymbol{\theta}}(\mathbf{x}, t)\|_2^2] \, dt + D_{\text{KL}}(p_T \parallel \pi)$$

shall be replaced with

$$D_{\text{KL}}(\tilde{p} \parallel \tilde{p}_{\boldsymbol{\theta}}^{\text{SDE}}) \leqslant \frac{1}{2} \int_\epsilon^T \mathbb{E}_{p_t(\mathbf{x})}[g(t)^2 \|\nabla_{\mathbf{x}} \log p_t(\mathbf{x}) - \boldsymbol{s}_{\boldsymbol{\theta}}(\mathbf{x}, t)\|_2^2] \, dt + D_{\text{KL}}(p_T \parallel \pi) \qquad (36)$$

where $\tilde{p}(\mathbf{x}) := \int p(\tilde{\mathbf{x}}) p_{0\epsilon}(\mathbf{x} \mid \tilde{\mathbf{x}}) \, d\mathbf{x}$, and $\tilde{p}_{\boldsymbol{\theta}}^{\text{SDE}}$ denotes the marginal distribution of $\hat{\mathbf{x}}_{\boldsymbol{\theta}}(\epsilon)$. Here the stochastic process $\{\hat{\mathbf{x}}_{\boldsymbol{\theta}}(t)\}_{t \in [0, T]}$ is defined according to Eq. (5). When $\epsilon$ is sufficiently small, we always have

$$D_{\text{KL}}(\tilde{p} \parallel \tilde{p}_{\boldsymbol{\theta}}^{\text{SDE}}) \approx D_{\text{KL}}(p \parallel p_{\boldsymbol{\theta}}^{\text{SDE}}),$$

so we train with Eq. (36) to approximately maximize the model likelihood for $p_{\boldsymbol{\theta}}^{\text{SDE}}$. However, at test time, we should report the likelihood bound for $p_{\boldsymbol{\theta}}^{\text{SDE}}$ for mathematical rigor, not $\tilde{p}_{\boldsymbol{\theta}}^{\text{SDE}}$. To this end, we first derive an analogous result to Theorem 3 with the time horizon $[\epsilon, T]$, given as below.

**Theorem 6.** *Let $p_{0t}(\mathbf{x}' \mid \mathbf{x})$ denote the transition distribution from $p_0(\mathbf{x})$ to $p_t(\mathbf{x})$ for the SDE in Eq. (1). With the same notations and conditions in Theorem 3, as well as the definitions of $\tilde{p}$ and $\tilde{p}_{\boldsymbol{\theta}}^{\text{SDE}}$ given above, we have*

$$-\mathbb{E}_{p_{0\epsilon}(\mathbf{x}'|\mathbf{x})}[\log \tilde{p}_{\boldsymbol{\theta}}^{\text{SDE}}(\mathbf{x}')] \leqslant \mathcal{L}_{\boldsymbol{\theta}}^{SM}(\mathbf{x}, \epsilon) = \mathcal{L}_{\boldsymbol{\theta}}^{DSM}(\mathbf{x}, \epsilon), \qquad (37)$$

*where $\mathcal{L}_{\boldsymbol{\theta}}^{SM}(\mathbf{x}, \epsilon)$ is defined as*

$$-\mathbb{E}_{p_{0T}(\mathbf{x}'|\mathbf{x})}[\log \pi(\mathbf{x}')] + \frac{1}{2} \int_\epsilon^T \mathbb{E}_{p_{0t}(\mathbf{x}'|\mathbf{x})} \left[ 2g(t)^2 \nabla_{\mathbf{x}'} \cdot \boldsymbol{s}_{\boldsymbol{\theta}}(\mathbf{x}', t) + g(t)^2 \|\boldsymbol{s}_{\boldsymbol{\theta}}(\mathbf{x}', t)\|_2^2 - 2\nabla_{\mathbf{x}'} \cdot \boldsymbol{f}(\mathbf{x}', t) \right] dt,$$

*and $\mathcal{L}_{\boldsymbol{\theta}}^{DSM}(\mathbf{x}, \epsilon)$ is given by*

$$-\mathbb{E}_{p_{0T}(\mathbf{x}'|\mathbf{x})}[\log \pi(\mathbf{x}')] + \frac{1}{2} \int_\epsilon^T \mathbb{E}_{p_{0t}(\mathbf{x}'|\mathbf{x})} \left[ g(t)^2 \|\boldsymbol{s}_{\boldsymbol{\theta}}(\mathbf{x}', t) - \nabla_{\mathbf{x}'} \log p_{0t}(\mathbf{x}' \mid \mathbf{x})\|_2^2 \right] dt$$
$$- \frac{1}{2} \int_\epsilon^T \mathbb{E}_{p_{0t}(\mathbf{x}'|\mathbf{x})} \left[ g(t)^2 \|\nabla_{\mathbf{x}'} \log p_{0t}(\mathbf{x}' \mid \mathbf{x})\|_2^2 + 2\nabla_{\mathbf{x}'} \cdot \boldsymbol{f}(\mathbf{x}', t) \right] dt.$$

*Proof.* The proof closely parallels that of Theorem 3, by noting that $\tilde{p}(\mathbf{x}) = \int p(\mathbf{x}') p_{0\epsilon}(\mathbf{x} \mid \mathbf{x}') \, d\mathbf{x}'$.
$\qquad\square$

Although $\tilde{p}_{\boldsymbol{\theta}}^{\text{SDE}}$ is a probabilistic model for $\tilde{p}$, we can transform it into a probabilistic model for $p$ leveraging a denoising distribution $q_{\boldsymbol{\theta}}(\mathbf{x} \mid \mathbf{x}')$ that approximately converts $\tilde{p}$ to $p$. Suppose $p_{0\epsilon}(\mathbf{x}' \mid \mathbf{x}) = \mathcal{N}(\mathbf{x}' \mid \alpha \mathbf{x}, \beta^2 \boldsymbol{I})$. Inspired by Tweedie's formula, we choose

$$q_{\boldsymbol{\theta}}(\mathbf{x} \mid \mathbf{x}') := \mathcal{N}\left( \mathbf{x} \;\middle|\; \frac{\mathbf{x}'}{\alpha} + \frac{\beta^2}{\alpha} \boldsymbol{s}_{\boldsymbol{\theta}}(\mathbf{x}', \epsilon), \frac{\beta^2}{\alpha^2} \boldsymbol{I} \right),$$

and define $p_{\boldsymbol{\theta}}(\mathbf{x}) := \int q_{\boldsymbol{\theta}}(\mathbf{x} \mid \mathbf{x}') \tilde{p}_{\boldsymbol{\theta}}^{\text{SDE}}(\mathbf{x}') \, \mathrm{d}\mathbf{x}'$, which is a probabilistic model for $p$. With slight abuse of notation, we identify $p_{\boldsymbol{\theta}}$ with $p_{\boldsymbol{\theta}}^{\text{SDE}}$ in Table 2. With Jensen's inequality, we have

$$-\log p_{\boldsymbol{\theta}}(\mathbf{x}) \leqslant -\mathbb{E}_{p_{0\epsilon}(\mathbf{x}'|\mathbf{x})}\left[\log \frac{q_{\boldsymbol{\theta}}(\mathbf{x} \mid \mathbf{x}') \tilde{p}_{\boldsymbol{\theta}}^{\text{SDE}}(\mathbf{x}')}{p_{0\epsilon}(\mathbf{x}' \mid \mathbf{x})}\right].$$

Combined with Theorem 6, we have

$$-\log p_{\boldsymbol{\theta}}(\mathbf{x}) \leqslant -\mathbb{E}_{p_{0\epsilon}(\mathbf{x}'|\mathbf{x})}[\log q_{\boldsymbol{\theta}}(\mathbf{x} \mid \mathbf{x}') - \log p_{0\epsilon}(\mathbf{x}' \mid \mathbf{x})] + \mathcal{L}_{\boldsymbol{\theta}}^{\text{SM}}(\mathbf{x}, \epsilon) \qquad (38)$$

$$= -\mathbb{E}_{p_{0\epsilon}(\mathbf{x}'|\mathbf{x})}[\log q_{\boldsymbol{\theta}}(\mathbf{x} \mid \mathbf{x}') - \log p_{0\epsilon}(\mathbf{x}' \mid \mathbf{x})] + \mathcal{L}_{\boldsymbol{\theta}}^{\text{DSM}}(\mathbf{x}, \epsilon) \qquad (39)$$

The above bound Eq. (39) was applied to both computing the test-time likelihood bounds in Table 2, and training the flow model used in variational dequantization. Note that it was not used to train the time-dependent score-based model.

In practice, we choose $\epsilon = 10^{-5}$ for VP SDEs and $\epsilon = 10^{-2}$ for subVP SDEs, except that on ImageNet we use $\epsilon = 5 \times 10^{-5}$ for VP SDE models trained with likelihood weighting and importance sampling. Note that [48] chooses $\epsilon = 10^{-5}$ for all cases. We found that when using likelihood weighting and optionally importance sampling, $\epsilon = 10^{-5}$ for subVP SDEs can cause stiffness for numerical ODE solvers. In contrast, using $\epsilon = 10^{-2}$ for subVP SDEs sidesteps numerical issues without hurting the performance for score-based models trained with original weightings in [48]. For the bound values in Table 2, we draw 1000 time values uniformly in $[\epsilon, T]$ and use them to estimate $\mathcal{L}_{\boldsymbol{\theta}}^{\text{DSM}}$ for each datapoint, with the same importance sampling technique in Eq. (12). We use the correction in Eq. (39) and report upper bounds for $-\log p_{\boldsymbol{\theta}}(\mathbf{x})$. For computing the likelihood of $p_{\boldsymbol{\theta}}^{\text{ODE}}$, we use the Dormand-Prince RK45 ODE solver [14] with absolute and relevant tolerances set to $10^{-5}$. We do not use the correction in Eq. (39) for $p_{\boldsymbol{\theta}}^{\text{ODE}}$, because it is still a valid likelihood for the data distribution even in the time horizon $[\epsilon, T]$.

Below is a related result to bound $\log \tilde{p}_{\boldsymbol{\theta}}^{\text{SDE}}(\mathbf{x})$ directly. We include it here for completeness, though we do not use it for either training or inference in our experiments.

**Theorem 7.** *Let $p_{0t}(\mathbf{x}' \mid \mathbf{x})$ denote the transition distribution from $p_0(\mathbf{x})$ to $p_t(\mathbf{x})$ for the SDE in Eq. (1). With the same notations and conditions in Theorem 3, as well as the definitions of $\tilde{p}$ and $\tilde{p}_{\boldsymbol{\theta}}^{\text{SDE}}$ in Theorem 6, we have*

$$-\log \tilde{p}_{\boldsymbol{\theta}}^{\text{SDE}}(\mathbf{x}) \leqslant \mathcal{L}_{\boldsymbol{\theta},\epsilon}^{SM}(\mathbf{x}) = \mathcal{L}_{\boldsymbol{\theta},\epsilon}^{DSM}(\mathbf{x}), \qquad (40)$$

*where $\mathcal{L}_{\boldsymbol{\theta},\epsilon}^{SM}(\mathbf{x})$ is defined as*

$$-\mathbb{E}_{p_{\epsilon T}(\mathbf{x}'|\mathbf{x})}[\log \pi(\mathbf{x}')] + \frac{1}{2}\int_\epsilon^T \mathbb{E}_{p_{\epsilon t}(\mathbf{x}'|\mathbf{x})}\left[2g(t)^2\nabla_{\mathbf{x}'} \cdot \boldsymbol{s}_{\boldsymbol{\theta}}(\mathbf{x}', t) + g(t)^2 \left\|\boldsymbol{s}_{\boldsymbol{\theta}}(\mathbf{x}', t)\right\|_2^2 - 2\nabla_{\mathbf{x}'} \cdot \boldsymbol{f}(\mathbf{x}', t)\right]\mathrm{d}t,$$

*and $\mathcal{L}_{\boldsymbol{\theta},\epsilon}^{DSM}(\mathbf{x})$ is given by*

$$-\mathbb{E}_{p_{\epsilon T}(\mathbf{x}'|\mathbf{x})}[\log \pi(\mathbf{x}')] + \frac{1}{2}\int_\epsilon^T \mathbb{E}_{p_{\epsilon t}(\mathbf{x}'|\mathbf{x})}\left[g(t)^2 \left\|\boldsymbol{s}_{\boldsymbol{\theta}}(\mathbf{x}', t) - \nabla_{\mathbf{x}'} \log p_{\epsilon t}(\mathbf{x}' \mid \mathbf{x})\right\|_2^2\right]\mathrm{d}t$$

$$-\frac{1}{2}\int_\epsilon^T \mathbb{E}_{p_{\epsilon t}(\mathbf{x}'|\mathbf{x})}\left[g(t)^2 \left\|\nabla_{\mathbf{x}'} \log p_{\epsilon t}(\mathbf{x}' \mid \mathbf{x})\right\|_2^2 + 2\nabla_{\mathbf{x}'} \cdot \boldsymbol{f}(\mathbf{x}', t)\right]\mathrm{d}t.$$

*Proof.* Proof closely parallels those of Theorems 3 and 6. $\qquad\square$

## C  Experimental details

**Datasets**  All our experiments are performed on two image datasets: CIFAR-10 [28] and down-sampled ImageNet [55]. Both contain images of resolution $32 \times 32$. CIFAR-10 has 50000 images as the training set and 10000 images as the test set. Down-sampled ImageNet has 1281149 training images and 49999 test images. It is well-known that ImageNet contains some personal sensitive information and may cause privacy concern [57]. We minimize this risk by using the dataset with a small resolution ($32 \times 32$).

**Model architectures**  Our variational dequantization model, $q_\phi(\mathbf{u} \mid \mathbf{x})$, follows the same architecture of Flow++ [18]. We do not use dropout for score-based models trained on ImageNet. We did not tune model architectures or training hyper-parameters specifically for maximizing likelihoods. All likelihood values were reported using the last checkpoint of each setting.

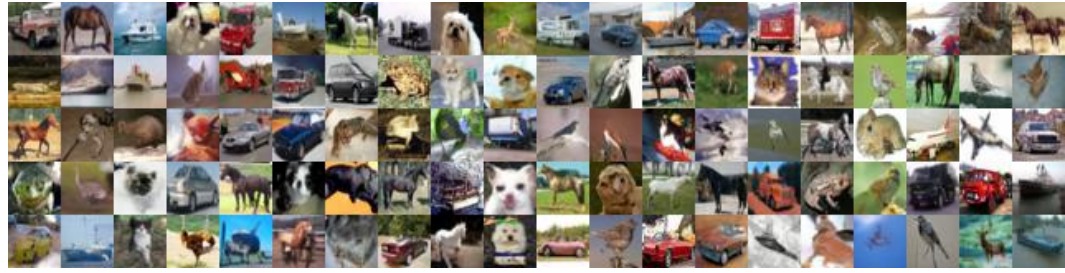

(a) DDPM++ (deep, subVP) [48], FID = 2.86

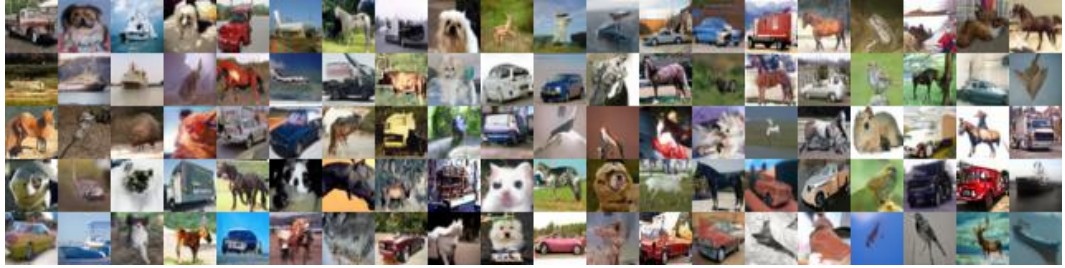

(b) ScoreFlow, FID = 5.34

Figure 3: Samples on CIFAR-10. (a) Model with the best FID. (b) ScoreFlow trained with likelihood weighting + importance sampling + VP SDE. Samples of both models are generated with the same random seed.

**Training**  We follow the same training procedure for score-based models in [48]. We also use the same hyperparameters for training the variational dequantization model, except that we train it for only 300000 iterations while fixing the score-based model. All models are trained on Cloud TPU v3-8 (roughly equivalent to 4 Tesla V100 GPUs). The baseline DDPM++ model requires around 33 hours to finish training, while the deep DDPM++ model requires around 44 hours. The variational dequantization model for the former requires around 7 hours to train, and for the latter it requires around 9.5 hours.

**Confidence intervals**  All likelihood values are obtained by averaging the results on around 50000 datapoints, sampled with replacement from the test dataset. We can compute the confidence intervals with Student's t-test. On CIFAR-10, the radius of 95% confidence intervals is typically around 0.006 bits/dim, while on ImageNet it is around 0.008 bits/dim.

**Sample quality**  All FID values are computed on 50000 samples from $p_\theta^{\text{ODE}}$, generated with numerical ODE solvers as in [48]. We compute FIDs between samples and training/test data for CIFAR-10/ImageNet. Although likelihood weighting + importance sampling slightly increases FID scores, their samples have comparable visual quality, as demonstrated in Figs. 3 and 4.

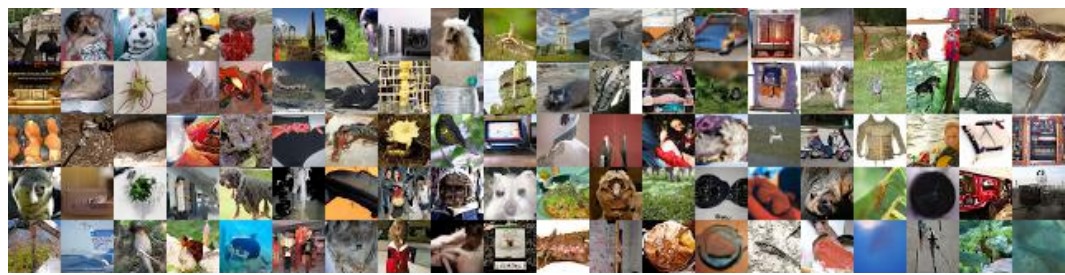

(a) DDPM++ (VP) [48], FID = 8.34

(b) ScoreFlow, FID = 10.18

Figure 4: Samples on ImageNet $32 \times 32$. (a) Model with the best FID. (b) ScoreFlow trained with likelihood weighting + importance sampling + VP SDE. Samples of both models are generated with the same random seed.