# OpenReview forum: "Maximum Likelihood Training of Score-Based Diffusion Models"
_NeurIPS.cc/2021/Conference — NeurIPS 2021 Spotlight_

### Official Review · Reviewer_3AeC · 2021-07-11

**Rating:** 7
**Confidence:** 4

**Summary:**

The paper studies score-based denoising-diffusion generative models and makes one major contribution. It shows how the training objective, a mixture or score-matching losses defined along different times of the diffusion, can be adapted to enable maximum likelihood training (using a lower bound objective). This is rigorously derived and proven analytically. The result is a specific weighting of the score-matching losses for different times t. The paper validates the derived methodology and empirically demonstrates very strong likelihood performance, state-of-the-art among models without data augmentations.

**Limitations And Societal Impact:**

Limitations and Societal Impact were not addressed in the paper in an optimal manner.

__Regarding limitations:__ I think obvious and well-known limitations of current score-based diffusion models, including the models and methods proposed in the paper, are:
1. The slow synthesis speed, since a deep network needs to be called very often to obtain the final results (by the ODE solver). This stands in contrast to models such as GANs or VAEs, from which one can sample very efficiently.
2. The used diffusion processes are defined for continuous data (or dequantized discrete data, if applicable, like for typical images), but cannot be used easily for truly discrete or categorical datatypes (e.g. language data, graphs, binary data, etc.).
3. As discussed, there seems to be some trade-off between strong FIDs and NLLs.

I think such limitations could be acknowledged and discussed better.

__Regarding Societal Impact:__ The authors state that they did not discuss societal impact, since the method per se does not add any new negative impacts. I basically agree with this statement. However, this work essentially tackles generative modeling, which indeed can have negative societal impacts, such as synthesizing fake images, etc. This is well-known, but I think a pointer in this direction to acknowledge this would not hurt.

**Main Review:**

__Overall impression:__ The main contribution of the paper, demonstrating how to train score-based diffusion models in a maximum likelihood fashion, using a lower bound objective, is a very important contribution. Score-based diffusion models recently generated much attention and demonstrated very high image quality synthesis, but their likelihoods were worse than those of some other models (autoregressive models, VAEs, etc.). This work solves this problem, demonstrates how to actually train the model directly towards good likelihood, and outperforms all previous models (when comparing to models without data augmentation). Consequently, I think this is a significant contribution and I expect future works on score-based diffusion models to often rely on the weightings derived in this paper. That being said, regarding the overall quality the paper lacks some discussions of important questions, and could discuss the context and limitations better. More specific feedback below.

__Clarity:__ I think the paper is well written and easily readable.

__Originality:__ The main contribution is original. The main contribution consists of several theorems that have not been appeared before, to the best of my knowledge. The proofs of the theorems seem solid and elegant. The paper also proposes importance sampling techniques and uses variational dequantization during training. However, these aspects are rather incremental.

__Weaknesses, Open Questions, Suggestions:__
1. The authors distinguish between $p^{ODE}$ and $p^{SDE}$, presenting it as if these are different generative models (e.g. in line 291 $p^{ODE}\neq p^{SDE}$). However, the authors also write that both $p^{ODE}$ and $p^{SDE}$ share the same marginal $p_t$ for all $t$ (line 117). Doesn't this imply that they define the same generative distribution and therefore model? Could we argue that $p^{ODE}$ and $p^{SDE}$ are merely two different ways of writing the *same* generative model, with one allowing us to sample with black-box ODE solvers, and the other, for example, with ancestral sampling (and differences in practice mainly being due to numerical issues and different solvers used)? With this interpretation it is also not surprising that the likelihoods/bounds of $p^{ODE}$ and $p^{SDE}$ improve together. Or is the issue that the statement "$p^{ODE}$ and $p^{SDE}$ share the same marginal $p_t$ for all $t$" is only true for valid score functions modeling proper distributions, while the learnt score function $\mathbf{s}_\theta$ may not satisfy these conditions? To me it rather feels the crucial difference is in the way they are evaluated - in one case via actual integration with error guarantees due to ODE solver tolerances, as in Normalizing flows, in the other case using a stochastic ELBO-like estimator. I got a bit confused and feel this point could be explained more clearly.
2. The models that achieve the strongest likelihoods tend to have worse FIDs than models trained without the likelihood weighting. There is a trade-off between NLL and FID. This is not discussed at all in the paper. How can we overcome this? However, looking at it from another perspective, the achieved FID scores, even for the models trained with the likelihood-weighting, are much better than typical FIDs obtained by previous autoregressive models, VAEs, flows, etc., which were also trained with likelihood objectives - and this is great! But why are FIDs so much better here? I think these aspects are quite interesting and important and deserve discussion.
3. One of the main drawbacks of current score-based diffusion models is their slow synthesis speed. This aspect is not discussed at all in the paper. What is the number of function evaluations, i.e. network calls, of the ODE solver when sampling? I think this needs to be reported and discussed as it is an important metric when working with diffusion models and can vary wildly. Ideally, even wall-clock time could be measured. It's also important when comparing to other types of likelihood-based models.
4. There is no thorough discussion of Related Works. I think contrasting the proposed *ScoreFlow* and its advantages and disadvantages to other types of likelihood-based generative modeling techniques and putting it into a broader context a bit more thoroughly would be nice.
5. The authors say they do not compare to the VESDE, since the VESDE likelihood weighting is the same as the weighting in [1]. It seems, however, that [1] did not evaluate likelihood for their models. Also, [1] obtained very strong FID score (although when using predictor-corrector sampling) even though they trained with a weighting corresponding to the likelihood-weighting derived in this work. This stands in contrast to the worse FID scores obtained here whenever using the likelihood weighting. I feel that exactly because the weighting used in [1] already is the maximum likelihood weighting, it would be very interesting to evaluate VESDE models towards likelihood or at least to discuss this.
6. Just a suggestion: If space is an issue for addressing some of my comments, I think the explanation of Variational Dequantization can be put in the Appendix entirely, as this is a standard technique. Also the proposed importance sampling is straight-forward and could be cut much shorter.

__Conclusion:__ In conclusion, I think this is a good paper that deserves to be published, as I think its main contribution is very important and it quantitative results are strong. However, I think the paper would significantly profit from addressing the above concerns and also considering the feedback regarding "Limitations and Societal Impact".

[1] Song et al. "Score-Based Generative Modeling through Stochastic Differential Equations"

**Time Spent Reviewing:**

5

---

> ### Author Response · Authors · 2021-08-10
> **Response**
>
> Thank you for the detailed review and thoughtful feedback. Below we address specific questions.
>
> ---------
>
> Q: Are $p^\text{ODE}$ and $p^\text{SDE}$ the same?
>
> A: The probability flow ODE indeed shares the same marginals $p_t$ as the SDE. However, when defining $p^\text{ODE}$ and $p^\text{SDE}$, we replace $\nabla_x \log p_t(x)$ with our learned score model $s_\theta(x, t)$, which results in approximate probability flow ODEs and reverse-time SDEs. This approximation error renders $p^\text{ODE}$ and $p^\text{SDE}$ different from each other. We will add more clarifications in Section 3.
>
> --------
>
> Q: Discussion on the trade-off between NLL and FID.
>
> A: As discussed by [1], better likelihood does not necessarily lead to better sample quality. Our goal in this paper is to improve the likelihood of score-based diffusion models. However, trying to improve the likelihood often causes degradation of FID scores. The same trend has been observed in [2] and [3] for discrete-time diffusion models as well. To trade NLL for FID, we can use weighting functions that interpolate between likelihood weighting and the original weighting functions. Our FID scores are much better than other likelihood-based models, since we build upon score-based diffusion models which are known to excel at FID scores. Understanding why score-based diffusion models have better FID scores than alternatives is an important direction for future research. We will include this discussion in Section 5.
>
> ----------
>
> Q: Reporting the number of function evaluations of the ODE solvers.
> A: We can include this info. Our ScoreFlow requires around 550 and 450 function evaluations on CIFAR-10 and ImageNet respectively. While slower than GANs or VAEs, this amounts to over a 5x speed-up compared to autoregressive models, which require 32x32x3 = 3072 function evaluations for 32 x 32 images. Since the number of function evaluations needed for sampling at higher resolutions does not tend to scale with dimensionality, this efficiency gap compared to autoregressive models only increases for higher-dimensional datasets.
>
> -------------
>
> Q: Discussion of related works.
>
> A: We are happy to include a more thorough discussion on related works using the additional 10-th content page, if our paper is accepted. On the other hand, a contribution of our paper is to show that score-based generative models, already classed as continuous normalizing flows in [4], can now be efficiently trained by maximum likelihood. In other words, score-based generative models trained by maximum likelihood are exactly continuous normalizing flows, but can be trained more efficiently since we don’t need to integrate across time for each training update. In this way, the context of our model is already established in the literature of normalizing flows.
>
> --------------
>
> Q: Likelihood achieved by VE SDE models.
>
> A: Although VE SDEs achieve very strong FID scores, we empirically found that their likelihood is inferior to VP or subVP SDEs. In fact, the best VE SDE model in [4] achieves an NLL around 3.4 bits/dim on CIFAR-10. We will clarify this around Table 1.
>
> ------------
>
> Q: Limitations and societal impact.
>
> A: Thanks for the detailed suggestions! As [reviewer CUCw](https://openreview.net/forum?id=AklttWFnxS9&noteId=z2i6lMX-A0S) pointed out, we have discussions about limits in Sec. 4 on Line 202-210 and Line 244-247. We will add a new section in the revision to summarize what we already have in the paper, as well as including your suggestions.
>
> ------------
>
> References
>
> [1] Theis, Lucas, Aäron van den Oord, and Matthias Bethge. "A note on the evaluation of generative models." arXiv preprint arXiv:1511.01844 (2015).
>
> [2] Ho, Jonathan, Ajay Jain, and Pieter Abbeel. "Denoising diffusion probabilistic models." arXiv preprint arXiv:2006.11239 (2020).
>
> [3] Nichol, Alex, and Prafulla Dhariwal. "Improved denoising diffusion probabilistic models." arXiv preprint arXiv:2102.09672 (2021).
>
> [4] Y. Song, J. Sohl-Dickstein, D. P. Kingma, A. Kumar, S. Ermon, and B. Poole. “Score-Based Generative Modeling Through Stochastic Differential Equations.” International Conference
> on Learning Representations, 2021.

---

### Official Review · Reviewer_gpyk · 2021-07-12

**Rating:** 8
**Confidence:** 5

**Summary:**

This paper shows that, with a specific weighting of the score matching loss, one can upper bound the negative log-likelihood of the density induced by the reverse SDE parameterized by the score function. The authors then propose to use importance sampling to minimize this weighted score matching loss and obtain state-of-the-art likelihood estimates on CIFAR10 and ImageNet 32x32.

**Limitations And Societal Impact:**

As pointed out in the main review above.

**Main Review:**

It is a strong submission with some theory and impressive empirical results, which furthermore bridges the long-standing gap between autoregressive models and non-autoregressive likelihood-based models.

In general, the paper is quite well written and quite easy to follow. Below I have some nits, which I hope the authors could address.

1. Thm 1 & 3: The main proof technique can be summarized by an application of data processing inequality (which upper bound marginal KL divergence with the joint, functional KL divergence) and Girsanov's theorem (for computing the functional KL). My main critique for Theorem 1 is that it does not hold for empirical distribution, as the differential entropy will be unbounded. Then the authors derive Theorem 3 to bound the likelihood of individual datum. However, I do not quite follow the step where the expectation E_p(x) is removed. Do you mean by taking p(x) to approximate a point mass centered at x? If so please make the statement rigorous. Also, I'm not sure why in the LHS of the equation after line 649, there is the conditional expectation p0t and x', is this a typo?

2. Thm 2: in the remark after the theorem, it's said that the densities induced by the SDE and ODE will be close if s\approx score of the data. There seems to be some logical leap here, since this will only hold true provided that the score of the model (\nabla log q, not s) approximates the score of the data (\nabla log q). It is still unclear how the two densities differ from or relate to each other, quantitatively and non-asymptotically (asymptotically it's clear when the score function s is perfect, then everything is fine), and why the ODE's likelihood is improved.

3. Bias in importance sampling: g^2/sigma^2 is not integrable as it blows up too fast as t->0, which means defining p(t) to be proportional to it will not result in a valid probability density. In Appendix B, it's said that the integration starts from some t_0, which should resolve the integrability issue, but this may result in bias in the ELBO estimation for both training and testing. Does the reported ELBO have this bias? I imagine this bias is negligible if t_0 is sufficiently small, has it been tested? Also, I'd suggest somehow include this in the main text so that people who want to build on the framework of this paper can be made aware of it.

4. VE SDE: Line 284, it is said that the weighting for VE SDE is the same as in [1]. But if I understand correctly, in [1], lambda is taken to be 1/E[nabla log p_0t], which will correspond to the "conditional" variance, not the diffusion coefficient sigma^2(t), aka likelihood weighting.
5. SOTA claim: It'd be better to include some other SOTA results with data augmentation in the table, such as [1,2]. It wouldn't harm the contribution of the paper as long as the SOTA claim is within the category of standard training with random horizontal flips.

[1] Score-Based Generative Modeling Through Stochastic Differential Equations.
[2] Distribution Augmentation for Generative Modeling
[3] Consistency Regularization for Variational Auto-Encoders

**Time Spent Reviewing:**

5

---

> ### Author Response · Authors · 2021-08-10
> **Response**
>
> Thank you for the detailed review and thoughtful feedback. Below we address specific questions.
>
> -----------
>
> Q: Theorem 1 does not hold for the empirical distribution. Dropping the $E_{p(x)}$ in Theorem 3. Typo in the LHS of the equation after line 649.
>
> A: You are right that Thm. 1 does not hold for the empirical distribution. As required by Assumption (i) in Appendix A, $p(x)$ needs to be a continuous distribution. Note that we use $p(x)$ to represent the underlying data distribution, not the empirical distribution.
>
> You have the right intuition about dropping the $E_{p(x)}$ in Thm. 3. If the bound in Thm. 3 does not hold for a particular data point x, we can always take $p(x)$ to be a continuous distribution centered around $x$. When the support of $p(x)$ is sufficiently small, the inequality in Thm. 5 will be violated due to continuity, which results in a contradiction. The proof is analogous to the proof for the fundamental lemma of calculus of variations. We are happy to provide a more detailed argument.
>
> Thanks for spotting the typo! The LHS should not contain $p_{0t}$ or $x’$. We will correct this in the revision.
>
> ---------
>
> Q: How the two densities differ from or relate to each other, quantitatively and non-asymptotically and why the ODE's likelihood is improved.
>
> A: We agree that Thm 2 does not fully address this question. As we stressed after Thm 2, there is no guarantee that improving the SDE likelihood will improve the ODE likelihood, and this is explicitly a limitation of our work. That said, our experimental results show that the ODE likelihood is always numerically bounded below by the SDE likelihood. We view it as an important future research direction to theoretically characterize the relation between these two densities.
>
> ---------
>
> Q: Bias in importance sampling and ELBO when integration starts from $t_0$.
>
> A: Great question! When integration starts from $t_0$, we are effectively modeling a noise-perturbed data distribution, where noise is added by running the SDE with the time horizon $[0, t_0]$. Therefore, this bias is negligible when $t_0$ is small. It is possible to correct for this bias using Jensen’s inequality, which we do in our updated results. We will include a discussion on this bias in the main text, using the additional 10-th page if our submission is accepted.
>
> ----------
>
> Q: Weightings in VE SDEs.
>
> A:  Previous applications of VE SDEs always use an exponentially growing schedule for $\sigma(t)$. In this case, the conditional variance is proportional to the squared diffusion coefficient. Therefore, the original weighting is effectively the same as likelihood weighting.
>
> -----------
>
> Q: Including results with data augmentation in the table.
>
> A: Thanks for the suggestion! We will compare with models using advanced data augmentation in Table 3.

---

> > ### Comment · Reviewer_gpyk · 2021-08-25
> > **Thanks for the update and response, still not clear about VE-SDE**
> >
> > Thank you for the response and the update on the empirical result. The authors addressed most of the questions I had, except for the equal weighting for VE-SDE.
> >
> > In fact, about VE-SDE, I think there might have been a mistake in [1]. While I tried to solve the VE-SDE as defined in eq (30) of [1], I didn't get (31) exactly. The variance in (31) should be
> >
> > $$\sigma_{\text{min}}^2 \left( \frac{\sigma_{\text{max}} }{\sigma_{\text{min}}}\right)^{2t} - \sigma_{\text{min}}^2$$
> >
> > This can be derived easily from the fact that $x_t$ following (30) is just a reparameterized Brownian motion, i.e. $x_t\overset{d}{=}w_{\int_0^t \sigma(t')^2 dt'}$, and the rest is just calculus.
> > This also makes more sense since now the variance will shrink to 0 when $t\rightarrow0$.
> >
> > That is, if we look at the correct "variance" of $p(x_t | x_0)$, it won't be the same as (or proportional to) $\sigma(t)^2$ due to the $-\sigma_\text{min}^2$ factor.
> >
> >
> >
> > [1] Y. Song, J. Sohl-Dickstein, D. P. Kingma, A. Kumar, S. Ermon, and B. Poole. “Score-Based Generative Modeling Through Stochastic Differential Equations.” International Conference on Learning Representations, 2021.

---

> > > ### Author Response · Authors · 2021-08-25
> > > **Further clarification about the VE-SDE**
> > >
> > > Thank you for the insightful question. As opposed to VP/subVP SDEs (eq. (32) of [1]), VE SDEs in [1] (eq. (30)) are only defined on the time horizon $(0, T]$, which doesn't contain $t=0$. From our understanding, $x_{0^+}$ is different from $x_0$ and is defined as a random variable from $\mathcal{N}(x_0, \sigma_{\text{min}}^2)$. By solving the VE SDE on $(0, T]$, we can obtain $$p(x_t \mid x_{0^+}) = \mathcal{N}\left( x_t \bigg| x_{0^+}, \sigma_{\text{min}}^2 \left( \frac{\sigma_{\text{max}}}{\sigma_{\text{min}}} \right)^{2t} - \sigma_{\text{min}}^2\right)$$
> > > Since $p(x_{0^+} \mid x_0) = \mathcal{N}(x_{0^+} \mid x_0, \sigma_{\text{min}}^2)$, we have
> > > $$
> > > p(x_t \mid x_0) = \mathcal{N}\left( x_t \bigg| x_0, \sigma_{\text{min}}^2 \left( \frac{\sigma_{\text{max}}}{\sigma_{\text{min}}} \right)^{2t} \right),
> > > $$
> > > which is the same as eq.(31) of [1]. This subtlety for $t=0$ is also briefly mentioned in Appendix C of [1], right below eq. (31). Moreover, it is the transition density used in the official implementation of [1] ([Github link](https://github.com/yang-song/score_sde/blob/4208a90823a06cba1fcc2d6b8974a94b37873f9e/sde_lib.py#L235)), so the weighting function in the official implementation coincides with our likelihood weighting.
> > >
> > > Reference:
> > >
> > > [1] Y. Song, J. Sohl-Dickstein, D. P. Kingma, A. Kumar, S. Ermon, and B. Poole. “Score-Based Generative Modeling Through Stochastic Differential Equations.” International Conference on Learning Representations, 2021.

---

### Official Review · Reviewer_CUCw · 2021-07-15

**Rating:** 7
**Confidence:** 3

**Summary:**

The authors show that for a specific weighting of score matching losses at various noise levels, the objective upper bounds the negative log-likelihood. They further bound the log-likelihood of individual datapoints to develop a continuous-time generalization of the ELBO.

Using the proposed objective the authors optimize score-based diffusion models towards likelihood.
The authors further make use of 1) importance sampling of time steps for reduced variance, 2) variational dequantization for a tighter bound on the likelihood and 3) the use of the equivalent ODE during evaluation, which allows exact likelihood computation (apart from the dequantization) for an approximation to the SDE model.
As a result of this, state-of-the-art likelihood performance is presented on CIFAR-10 and ImageNet 32x32, beating even the best autoregressive models.

**Limitations And Societal Impact:**

Some limitations of the work are discussed in Sec. 4 on Line 202-210 and Line 244-247.
Potential societal impacts of the work are not discussed.

**Main Review:**

Explicitly outlining the connections to maximum likelihood estimation (MLE) for continuous-time score-based diffusion models is novel, concurrent with [1], which was an important missing piece. It is nice to see a continuous-time extension of the ELBO. One thing I found missing was a connection to the discrete-time case as in DDPM [2] where a link between the ELBO and the score-matching loss is also established. I feel a discussion on this would clarify connections and thus strengthen the paper.

As for the results, the ablation study in Table 2 is informative, showing that
1. likelihood weighting tends to improve likelihood, empirically backing up the choice, while degrading the FID score.
1. the performance of the equivalent ODE tends to perform better than the SDE despite this not being guaranteed.
1. the proposed importance sampling improves performance, along with the use of variational dequantization and depth.

Furthermore, the best numbers reported, as summarized in Table 3, are quite impressive, setting new records on two highly benchmarked datasets, which have been dominated by autoregressive models for years. This demonstrates clearly that score-based diffusion models have great potential, not just for image synthesis, but also for tasks relying on the likelihood (e.g. compression).

As for the clarity: I believe parts of Section 4 could be compressed and simplified to improve clarity. Please correct me if I'm wrong.
Thm. 1 seems superfluous given Thm. 2 which turns the inequality into an equality (obtained through Eq. 25-27 with $q:=p_{\theta}^{\text{SDE}}$). Also, for Thm. 2, it seems $q_t$ is defined as the process solving Eq. (5) and thus $q := q_0 := p_{\theta}^{\text{SDE}}$ trivially, so it seems $q$ could be dropped?
Finally, would not the requirement for $p_{\theta}^{\text{ODE}} = p_{\theta}^{\text{SDE}}$ be that $s_{\theta}(x,t) = \nabla_x \log p_t(x)$ instead of $\nabla_x \log q_t(x)$?

Minor fix on Line 127: $s_{\theta}(x) \rightarrow s_{\theta}(x, t)$.

[1] A Variational Perspective on Diffusion-Based Generative Models and Score Matching (Huang et al., 2021)
[2] Denoising Diffusion Probabilistic Models (Ho et al., 2020)

**Time Spent Reviewing:**

6

---

> ### Author Response · Authors · 2021-08-10
> **Response**
>
> Thank you for the detailed review and thoughtful feedback. Below we address specific questions.
>
> -----------------
>
> Q: A discussion on the connection to the discrete-time case as in DDPM.
>
> A: Thanks for the suggestion! We mentioned in line 237-238 that our bound can be viewed as a continuous-time generalization of the ELBO in DDPM. We are also happy to include further discussion.
>
> ---------------
>
> Q: Thm. 1 seems superfluous given Thm. 2. Clarifications on the conditions in Thm. 2.
>
> A: The equality in Thm. 2 holds only when $s_\theta(x, t) = \nabla_x \log p_\theta^\text{SDE}(x)$ (i.e. our model corresponds to the score of some well-defined density), which is a strong requirement and is often violated in practice (see line 202-204). When this condition does not hold, the equality in Thm. 2 becomes the inequality in Thm. 1. Therefore, Thm. 1 is not implied by Thm. 2.
>
> $q_t$ is defined as the process solving eq. (1) or equivalently eq. (2). It is not the same as the process given by eq. (5) unless $s_\theta(x, t) = \nabla_x \log q_t(x)$.
>
> In Thm. 2, the noise-perturbed data distribution $p_t$ and model distribution $q_t$ can be different, and we only require that $s_\theta(x) = \nabla_x \log q_t(x)$ for the equality to hold. This indicates that $s_\theta(x)$ may not equal $\nabla_x \log p_t(x)$.
>
> We are happy to add more clarifications after the statement of Thm. 2.

---

> > ### Comment · Reviewer_CUCw · 2021-08-18
> > **Thanks**
> >
> > Thanks for the clarifications.
> > This paper adds to our knowledge of the promising class of score-based diffusion models.
> > With the promised additions, I'm thus happy to accept this paper.

---

### Official Review · Reviewer_Vbrn · 2021-07-17

**Rating:** 7
**Confidence:** 4

**Summary:**

This paper presents likelihood weighting to improve the log-likelihood performance of the score-based model. It is widely known that the score-based model is an excellent generative model to sample an instance from the high-dimensional data distribution. However, its encoding capability has remained question. This paper argues that the presented likelihood weighting can be universally applicable to the score-based models to improve such encoding capability without hurting the generation performance. There were series of theoretic derivations to induce the likelihood weighting, and a benchmark experiment was conducted.


**Limitations And Societal Impact:**


There is no discussion at all. I understand that authors may not discuss the societal impact, but they can surely include the limitations that they expect from their models.




**Main Review:**

1)
The main question is improving negative log-likelihood without damaging FID. Table 2 shows that "DEEP+LW+IS" achieves the best NLL with mediocre FID because the best NLL case results in 5.70 FID when the best FID is 2.86. When we observe Table 2, FID score almost doubles whenever we introduce likelihood weighting. There is no discussion about FID in Table 2 except "with only slight degradation of sample quality as measured by FID" in line 299.

Then, these questions arise.
- Why do you need additional importance sampling to limit the damage on FID? What is the theory behind of this observation?
- Still, the degradation of the FID is notieable. Can you verify this sample quality in other metrics, i.e. IS?
- Is there any trade-off coefficient that you can introduce to barter NLL with FID?


2)
Why do you report only NLL in Table 3 without FID? To empirically verify your claim, don't you need to observe the two aspects at the same time? Or, is it because your score model has definately better FID over the flow, transformer, and other models?


3)
Likelihood weighting seems to increase the loss variance as you mentioned in line 252. It seems that importance weighting is introduced to limit this side effect in an adhoc manner. To reason thie phenomena further, there should be inequality checking of loss variances between the case of using $\lambda$ as before and $\lambda=g(t)^2$.

4)
VE-SDE uses the same weighting of likelihood weighting and the weighting in the prior work. Then, people may just use the VE-SDE variation of score-based models to optimize the NLL.



**Time Spent Reviewing:**

3

---

> ### Author Response · Authors · 2021-08-10
> **Response**
>
> Thank you for the detailed review and thoughtful feedback. Below we address specific questions.
>
> ----------------
>
> Q: Likelihood weighting increases FID scores.
>
> A: With likelihood weighting, we aim to further improve the likelihood of score-based diffusion models for potential applications such as lossless compression—we do not aim to improve FID scores, and reporting how FID changes for models trained to maximize likelihood is itself a contribution of this paper. We also note that better likelihood does not necessarily lead to better sample quality [1]. In fact, as observed in [2][3], likelihood-favoring objectives often result in worse FID scores for discrete-time diffusion models.
>
> ------------------
>
> Q: Degradation of FID is noticeable. Can you verify this sample quality in other metrics, i.e. IS?
>
> A: The reviewer correctly noticed that with likelihood weighting, FID scores nearly doubled on CIFAR-10. Note that this is not true on ImageNet, and only happens on CIFAR-10 because FID scores there are already very small. In fact, FIDs only increase by around 2.5 for both CIFAR-10 and ImageNet in Table 2. This small degradation of FID scores causes very mild differences in sample quality. Please see Figure 3 & 4 in the supplementary material to check samples before and after likelihood weighting. Aside from line 299, we will add more discussions in Section 5.1, and mention Figure 3 & 4 explicitly in the main text.
>
> Inception scores (IS) have the same trend as FID scores in our case. We omitted IS because it has many well-documented defects [4], but we are happy to include it if the reviewers/AC think it can improve the paper.
>
> ----------
>
> ​​Q: Why do you need additional importance sampling to limit the damage on FID? What is the theory behind this observation?
>
> A: Importance sampling can reduce the variance of our training objective (see Figure 2) and improve optimization. With better optimization, it would be expected that the score model has higher performance in both FIDs and NLLs, as can be observed in Table 2 by comparing models with LW and LW + IS.
>
> ---------
>
> Q: Is there any trade-off coefficient that you can introduce to barter NLL with FID?
>
> A: Since the original weightings in Table 1 have better FID scores but worse NLLs compared to our likelihood weighting, interpolating between those two weighting functions can trade NLL for FID.
>
> ---------
>
> Q: Why do you report only NLL in Table 3 without FID? To empirically verify your claim, don't you need to observe the two aspects at the same time? Or, is it because your score model has definitely better FID over the flow, transformer, and other models?
>
> A: The FIDs of ScoreFlow can be found in Table 2. We are happy to include FID scores for other models if the reviewer and AC thinks it can improve the paper. We emphasize our claim that likelihood weighting improves likelihood values of score-based models with only slight degradation of FID scores, which is verified by results in Table 2. Our model indeed has better FID scores than existing flow models and autoregressive models. For example, our ScoreFlow on CIFAR-10 has an FID of 5.7, whereas the SOTA autoregressive model, Sparse Transformer, has an FID over 27.
>
> ------------
>
> Q: There should be inequality checking of loss variances before and after importance sampling.
>
> A: We compared the variances of learning curves in Figure 2 and included a discussion in Section 5.1. From the “wiggliness” of curves in Figure 2 it should be clear that importance sampling significantly reduces the loss variance. In fact, with importance sampling, the loss variance (after convergence) decreases from 98.48 to 0.068 on CIFAR-10, and decreases from 0.51 to 0.043 on ImageNet. We will include these numbers in the revision.
>
> ---------
>
> Q: VE-SDE uses the same weighting of likelihood weighting and the weighting in the prior work. Then, people may just use the VE-SDE variation of score-based models to optimize the NLL.
>
> A: Great question! Even with the same likelihood weighting, different SDEs can achieve different NLLs (e.g., see VP vs. subVP in Table 2). We found that the best VE SDE model in [5] achieves an NLL around 3.4 bits/dim on CIFAR-10. It is not competitive against VP & subVP SDEs, whose likelihood weightings were unknown before our work. We will clarify this around Table 1.
>
> ---------
>
> Q: No discussion about limits.
>
> A: As [reviewer CUCw](https://openreview.net/forum?id=AklttWFnxS9&noteId=z2i6lMX-A0S) pointed out, we have discussions about limits in Sec. 4 on Line 202-210 and Line 244-247. We will also incorporate the suggestions of [reviewer 3AeC](https://openreview.net/forum?id=AklttWFnxS9&noteId=8aN0mRUcFsl) by adding a new section in the revision for more detailed discussion on limits and negative societal impact.
>
> ---------
>
> References
>
> [1] Theis, Lucas, Aäron van den Oord, and Matthias Bethge. "A note on the evaluation of generative models." arXiv preprint arXiv:1511.01844 (2015).
>
> [2] Ho, Jonathan, Ajay Jain, and Pieter Abbeel. "Denoising diffusion probabilistic models." arXiv preprint arXiv:2006.11239 (2020).
>
> [3] Nichol, Alex, and Prafulla Dhariwal. "Improved denoising diffusion probabilistic models." arXiv preprint arXiv:2102.09672 (2021).
>
> [4] ​​Barratt, Shane, and Rishi Sharma. "A note on the inception score." arXiv preprint arXiv:1801.01973 (2018).
>
> [5] Y. Song, J. Sohl-Dickstein, D. P. Kingma, A. Kumar, S. Ermon, and B. Poole. “Score-Based Generative Modeling Through Stochastic Differential Equations.” International Conference
> on Learning Representations, 2021.

---

> > ### Comment · Reviewer_Vbrn · 2021-08-10
> > **Thanks for the info**
> >
> > Authors comment resolves most of my questions, and I do not have any further.
> >
> > The paper is nicely written with many interesting theoretic knowledge.
> > I will increase the score to be a good paper
> >
> > Thanks

---

### Author Response · Authors · 2021-08-10
**Updated results**

We would like to thank all reviewers for providing high quality reviews and constructive feedback. After the paper submission, we identified two implementation issues that affected our results in Table 2:

1. Float32 causes numerical instability when computing the SDE’s drift/diffusion coefficients and transition densities for $t \approx 0$. We have fixed this issue by applying Taylor expansion at small $t$, and verified that it performs comparably to float64.
2. A bug in variational dequantization causes slight overestimation of the likelihoods. We have fixed this bug in the updated results.

Aside from fixing these issues, we also improved the variational dequantization network with a full Flow++ model. We re-trained and re-evaluated all models, and updated the results in Table 2. Here is the updated table:

|       |        |     |     |   CIFAR-10 |    |     |   |    | ImageNet |    |   |
|:----:|:----:|:------:|:------:|:-----:|:-----:|:-----:|:-----:|:-----:|:------:|:----:|:----:|
| Model    |   SDE  |  NLL (Uni. deq.) | Bound (Uni. deq.) | NLL  (Var. deq.) | Bound (Var. deq.) | FID | NLL (Uni. deq.) | Bound (Uni. deq.) | NLL (Var. deq.) | Bound (Var. deq.) | FID |
| Baseline | VP | 3.16 | 3.28 | 3.04 | 3.14 | 3.98 | 3.90 | 3.96 | 3.84 | 3.91 | **8.34** |
| Baseline + LW | VP | 3.06 | 3.18 | 2.94 | 3.03 | 5.18 | 3.91 | 3.96 | 3.86 | 3.92 | 17.75 |
| Baseline + LW + IS | VP | 2.95 | 3.08 | 2.83 | 2.94 | 6.03 | 3.86 | 3.92 | 3.80 | 3.88 | 11.15 |
| Deep | VP | 3.13 | 3.25 | 3.01 | 3.10 | 3.09 | 3.89 | 3.95 | 3.84 | 3.90 | 8.40 |
| Deep + LW | VP | 3.06 | 3.17 | 2.93 | 3.02 | 7.88 | 3.91 | 3.96 | 3.86 | 3.92 | 17.73 |
| Deep + LW + IS | VP | 2.93 | 3.06 | **2.80** | 2.92 | 5.34 | 3.85 | 3.92 | 3.79 | 3.88 | 11.20 |
| Baseline | subVP | 2.99 | 3.09 | 2.88 | 2.98 | 3.20 | 3.87 | 3.92 | 3.82 | 3.88 | 8.71 |
| Baseline + LW | subVP | 2.97 | 3.07 | 2.86 | 2.96 | 7.33 | 3.87 | 3.92 | 3.82 | 3.88 | 12.99 |
| Baseline + LW + IS | subVP | 2.94 | 3.05 | 2.84 | 2.94 | 5.58 | 3.84 | 3.91 | 3.79 | 3.87 | 10.57 |
| Deep | subVP | 2.96 | 3.06 | 2.85 | 2.95 | **2.86** | 3.86 | 3.91 | 3.81 | 3.87 | 8.87 |
| Deep + LW | subVP | 2.95 | 3.05 | 2.85 | 2.94 | 6.57 | 3.88 | 3.93 | 3.83 | 3.88 | 16.55 |
| Deep + LW + IS | subVP | **2.90** | **3.02** | 2.81 | **2.90** | 5.40 | **3.82** | **3.90** | **3.76** | **3.86** | 10.18 |


**Importantly: the central claims in our paper are still valid, except that the NLL of our best model on CIFAR-10 degrades from 2.74 bits/dim to 2.80 bits/dim.** Since it now matches (not outperforms) the former SOTA model Sparse Transformer, we retract the SOTA claim on CIFAR-10. On ImageNet 32x32, our best model keeps the same NLL of 3.76 bits/dim, still better than SOTA at time of submission.

---

### Decision · Program_Chairs · 2021-09-27

**Decision:**

Accept (Spotlight)

**Comment:**

This paper shows how training continuous-time denoising diffusion models (a.k.a. score-based modes) can be converted to maximum-likelihood training with a proper weighting of the training objective. Although this connection is known to some degrees from classical literature (see below), the paper does an excellent job of showing how following this connection for training yields "SOTA" likelihood results with modern score-based generative models and proper importance sampling. Given the unanimously positive reviews and the potential impact of this work, I am very happy to recommend this paper for acceptance.

That said, I'd like to encourage the authors to make the following changes for the camera-ready version:
- I appreciate coming forward with the updated numbers after discovering the numerical problems. Please make sure that the claims in this submission are adjusted after updating the numbers in the final camera-ready version.
- The connection between the KL divergence and score matching objective was also discussed in classical papers such as [1]. Please add a small section discussing these earlier works.

[1] Lyu, Interpretation and Generalization of Score Matching.